# Generating Multiple Objects at Spatially Distinct Locations

**Tobias Hinz, Stefan Heinrich, Stefan Wermter**
Knowledge Technology, Department of Informatics, Universität Hamburg
Vogt-Koelln-Str. 30, 22527 Hamburg, Germany
https://www.inf.uni-hamburg.de/en/inst/ab/wtm/
`{hinz,heinrich,wermter}@informatik.uni-hamburg.de`

## Abstract

Recent improvements to Generative Adversarial Networks (GANs) have made it possible to generate realistic images in high resolution based on natural language descriptions such as image captions. However, fine-grained control of the image layout, i.e. where in the image specific objects should be located, is still difficult to achieve. We introduce a new approach which allows us to control the location of arbitrarily many objects within an image by adding an object pathway to both the generator and the discriminator. Our approach does not need a detailed semantic layout but only bounding boxes and the respective labels of the desired objects are needed. The object pathway focuses solely on the individual objects and is iteratively applied at the locations specified by the bounding boxes. The global pathway focuses on the image background and the general image layout. We perform experiments on the Multi-MNIST, CLEVR, and the more complex MS-COCO data set. Our experiments show that through the use of the object pathway we can control object locations within images and can model complex scenes with multiple objects at various locations. We further show that the object pathway focuses on the individual objects and learns features relevant for these, while the global pathway focuses on global image characteristics and the image background.

## 1 Introduction

Understanding how to learn powerful representations from complex distributions is the intriguing goal behind adversarial training on image data. While recent advances have enabled us to generate high-resolution images with Generative Adversarial Networks (GANs), currently most GAN models still focus on modeling images that either contain only one centralized object (e.g. faces (CelebA), objects (ImageNet), birds (CUB-200), flowers (Oxford-102), etc.) or on images from one specific domain (e.g. LSUN bedrooms, LSUN churches, etc.). This means that, overall, the variance between images used for training GANs tends to be low (Raj et al., 2017). However, many real-life images contain multiple distinct objects at different locations within the image and with different relations to each other. This is for example visible in the MS-COCO data set (Lin et al., 2014), which consists of images of different objects at different locations within one image. In order to model images with these complex relationships, we need models that can model images containing multiple objects at distinct locations. To achieve this, we need control over what kind of objects are generated (e.g. persons, animals, objects, etc.), the location, and the size of these objects. This is a much more challenging task than generating a single object in the center of an image.

Current work (Karacan et al., 2016; Johnson et al., 2018; Hong et al., 2018b; Wang et al., 2018) often approaches this challenge by using a semantic layout as additional conditional input. While this can be successful in controlling the image layout and object placement, it also places a high burden on the generating process since a complete scene layout must be obtained first. We propose a model that does not require a full semantic layout, but instead only requires the desired object locations and identities (see Figure 1). One part of our model, called the global pathway, is responsible for generating the general layout of the complete image, while a second path, the object pathway, is used to explicitly generate the features of different objects based on the relevant object label and location.

The generator gets as input a natural language description of the scene (if existent), the locations and labels of the various objects within the scene, and a random noise vector. The global pathway uses this to create a scene layout encoding which describes high-level features and generates a global feature representation from this. The object pathway generates a feature representation of a given object at a location described by the respective bounding box and is applied iteratively over the scene at the locations specified by the individual bounding boxes. We then concatenate the feature representations of the global and the object pathway and use this to generate the final image.

The discriminator, which also consists of a global and object pathway, gets as input the image, the bounding boxes and their respective object labels, and the textual description. The global pathway is then applied to the whole image and obtains a feature representation of the global image features. In parallel, the object pathway focuses only on the areas described by the bounding boxes and the respective object labels and obtains feature representations of these specific locations. Again, the outputs of both the global and the object pathway are merged and the discriminator is trained to distinguish between real and generated images.

In contrast to previous work we do not generate a scene layout of the whole scene but only focus on relevant objects which are placed at the specified locations, while the global consistency of the image is the responsibility of the other part of our model. To summarize our model and contributions: 1) We propose a GAN model that enables us to control the layout of a scene without the use of a scene layout. 2) Through the use of an object pathway which is responsible for learning features of different object categories, we gain control over the identity and location of arbitrarily many objects within a scene. 3) The discriminator judges not only if the image is realistic and aligned to the natural language description, but also whether the specified objects are at the given locations and of the correct object category. 4) We show that the object pathway does indeed learn relevant features for the different objects, while the global pathway focuses on general image features and the background.

## 2 RELATED WORK

Having more control over the general image layout can lead to a higher quality of images (Reed et al., 2016a; Hong et al., 2018b) and is also an important requirement for semantic image manipulation (Hong et al., 2018a; Wang et al., 2018). Approaches that try to exert some control over the image layout utilize Generative Adversarial Nets (Goodfellow et al., 2014), Refinement Networks (e.g. Chen & Koltun (2017); Xu et al. (2018a)), recurrent attention-based models (e.g. Mansimov et al. (2016)), autoregressive models (e.g. Reed et al. (2016c)), and even memory networks supplying the image generation process with previously extracted image features (Zhang et al., 2018b).

One way to exert control over the image layout is by using natural language descriptions of the image, e.g. image captions, as shown by Reed et al. (2016b), Zhang et al. (2018a), Sharma et al. (2018), and Xu et al. (2018b). However, these approaches are trained only with images and their respective captions and it is not possible to specifically control the layout or placement of specific objects within the image. Several approaches suggested using a semantic layout of the image, generated from the image caption, to gain more fine-grained control over the final image. Karacan et al. (2016), Johnson et al. (2018), and Wang et al. (2018) use a scene layout to generate images in which given objects are drawn within their specified segments based on the generated scene layout. Hong et al. (2018b) use the image caption to generate bounding boxes of specific objects within the image and predict the object's shape within each bounding box. This is further extended by Hong et al. (2018a) by making it possible to manipulate images on a semantic level. While these approaches offer a more detailed control over the image layout they heavily rely on a semantic scene layout for the image generating process, often implying complex preprocessing steps in which the scene layout is constructed.

The two approaches most closely related to ours are by Reed et al. (2016a) and Raj et al. (2017). Raj et al. (2017) introduce a model that consists of individual "blocks" which are responsible for different object characteristics (e.g. color, shape, etc.). However, their approach was only tested on the synthetic SHAPES data set (Andreas et al., 2016), which has only comparatively low variability and no image captions. Reed et al. (2016b) condition both the generator and the discriminator on either a bounding box containing the object or keypoints describing the object's shape. However, the used images are still of relatively low variability (e.g. birds (Wah et al., 2011)) and only contain one object, usually located in the center of the image. In contrast, we model images with several different objects at various locations and apply our object pathway multiple times at each image, both in the

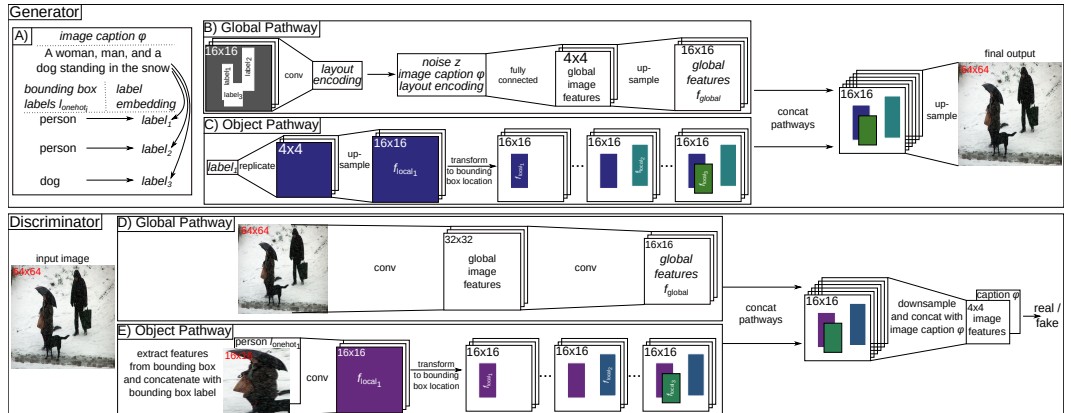

Figure 1: Both the generator and the discriminator of our model consist of a global and an object pathway. The global pathway focuses on global image characteristics, such as the background, while the object pathway is responsible for modeling individual objects at their specified location.

generator and in the discriminator. Additionally, we use the image caption and bounding box label to obtain individual labels for each bounding box, while Reed et al. (2016b) only use the image caption as conditional information.

## 3 APPROACH

For our approach[1], the central goal is to generate objects at arbitrary locations within a scene while keeping the scene overall consistent. For this we make use of a generative adversarial network (GAN) (Goodfellow et al., 2014). A GAN consists of two networks, a generator and a discriminator, where the generator tries to reproduce the true data distribution and the discriminator tries to distinguish between generated data points and data points sampled from the true distribution. We use the conditional GAN framework, in which both the generator and the discriminator get additional information, such as labels, as input. The generator $G$ (see Figure 1) gets as input a randomly sampled noise vector $z$, the location and size of the individual bounding boxes $bbox_i$, a label for each of the bounding boxes encoded as a one-hot vector $l_{\mathrm{onehot}_i}$, and, if existent, an image caption embedding $\varphi$ obtained with a pretrained char-CNN-RNN network from Reed et al. (2016b). As a pre-processing step (A), the generator constructs labels $label_i$ for the individual bounding boxes from the image caption $\varphi$ and the provided labels $l_{\mathrm{onehot}_i}$ of each bounding box. For this, we concatenate the image caption embedding $\varphi$ and the one-hot vector of a given bounding box $l_{\mathrm{onehot}_i}$ and create a new label embedding $label_i$ by applying a matrix-multiplication followed by a non-linearity (i.e. a fully connected layer). The resulting label $label_i$ contains the previous label as well as additional information from the image caption, such as color or shape, and is potentially more meaningful. In case of missing image captions, we use the one-hot embedding $l_{\mathrm{onehot}_i}$ only.

The generator consists of two different streams which get combined later in the process. First, the *global pathway* (B) is responsible for creating a general layout of the global scene. It processes the previously generated local labels $label_i$ for each of the bounding boxes and replicates them spatially at the location of each bounding box. In areas where the bounding boxes overlap the label embeddings $label_i$ are summed up, while the areas with no bounding boxes remain filled with zeros. Convolutional layers are applied to this layout to obtain a high-level layout encoding which is concatenated with the noise vector $z$ and the image caption embedding $\varphi$ and the result is used to generate a general image layout $f_{\mathrm{global}}$.

Second, the *object pathway* (C) is responsible for generating features of the objects $f_{\mathrm{local}_i}$ within the given bounding boxes. This pathway creates a feature map of a predefined resolution using convolutional layers which receive the previously generated label $label_i$ as input. This feature map is further transformed with a Spatial Transformer Network (STN) (Jaderberg et al., 2015) to fit into the

---

[1]Code can be found here: `https://github.com/tohinz/multiple-objects-gan`

bounding box at the given location on an empty canvas. The same convolutional layers are applied to each of the provided labels, i.e. we have one object pathway that is applied several times across different labels $label_i$ and whose output feeds onto the corresponding coordinates on the empty canvas. Again, features within overlapping bounding box areas are summed up, while areas outside of any bounding box remain zero.

As a final step, the outputs of the global and object pathways $f_{\text{global}}$ and $f_{\text{local}_i}$ are concatenated along the channel axis and are used to generate the image in the final resolution, using common GAN procedures. The specific changes of the generator compared to standard architectures are the object pathway that generates additional features at specific locations based on provided labels, as well as the layout encoding which is used as additional input to the global pathway. These two extensions can be added to the generator in any existing architecture with limited extra effort.

The discriminator receives as input an image (either original or generated), the location and size of the bounding boxes $bbox_i$, the labels for the bounding boxes as one-hot vectors $l_{\text{onehot}_i}$, and, if existent, the image caption embedding $\varphi$. Similarly to the generator, the discriminator also possesses both a *global* (D) and an *object* (E) pathway respectively. The global pathway takes the image and applies multiple convolutional layers to obtain a representation $f_{\text{global}}$ of the whole image. The object pathway first uses a STN to extract the objects from within the given bounding boxes and then concatenates these extracted features with the spatially replicated bounding box label $l_{\text{onehot}_i}$. Next, convolutional layers are applied and the resulting features $f_{\text{local}_i}$ are again added onto an empty canvas within the coordinates specified by the bounding box. Note, similarly to the generator we only use one object pathway that is applied to multiple image locations, where the outputs are then added onto the empty canvas, summing up overlapping parts and keeping areas outside of the bounding boxes set to zero. Finally, the outputs of both the object and global pathways $f_{\text{local}_i}$ and $f_{\text{global}}$ are concatenated along the channel axis and we again apply convolutional layers to obtain a merged feature representation. At this point, the features are concatenated either with the spatially replicated image caption embedding $\varphi$ (if existent) or the sum of all one-hot vectors $l_{\text{onehot}_i}$ along the channel axis, one more convolutional layer is applied, and the output is classified as either generated or real.

For the general training, we can utilize the same procedure that is used in the GAN architecture that is modified with our proposed approach. In our work we mostly use the StackGAN (Zhang et al., 2018a) and AttnGAN (Xu et al., 2018b) frameworks which use a modified objective function taking into consideration the additional conditional information and provided image captions. As such, our discriminator $D$ and our generator $G$ optimize the following objective function:

$$\min_G \max_D V(D, G) = \mathbb{E}_{(x,c)\sim p_{\text{data}}}[log D(x, c)] + \mathbb{E}_{(z)\sim p_z, (c)\sim p_{\text{data}}}[log(1 - D(G(z, c), c))],$$

where $x$ is an image, $c$ is the conditional information for this image (e.g. $label_i$, bounding boxes $bbox_i$, or an image caption $\varphi$), $z$ is a randomly sampled noise vector used as input for $G$, and $p_{\text{data}}$ is the true data distribution. Zhang et al. (2018a) and others use an additional technique called conditioning augmentation for the image captions which helps improve the training process and the quality of the generated images. In the experiments in which we use image captions (MS-COCO) we also make use of this technique[2].

## 4 EVALUATION AND ANALYSIS

For the evaluation, we aim to study the quality of the generated images with a particular focus on the generalization capabilities and the contribution of specific parts of our model, in both controllable and large-scale cases. Thus, in the following sections, we evaluate our approach on three different data sets: the Multi-MNIST data set, the CLEVR data set, and the MS-COCO data set.

### 4.1 MULTI-MNIST

In our first experiment, we used the Multi-MNIST data set (Eslami et al., 2016) for testing the basic functionality of our proposed model. Using the implementation provided by Eslami et al. (2016), we created 50,000 images of resolution $64 \times 64$ px that contain exactly three normal-sized MNIST digits in non-overlapping locations on a black background.

---

[2]More detailed information about the implementation can be found in the Appendix.

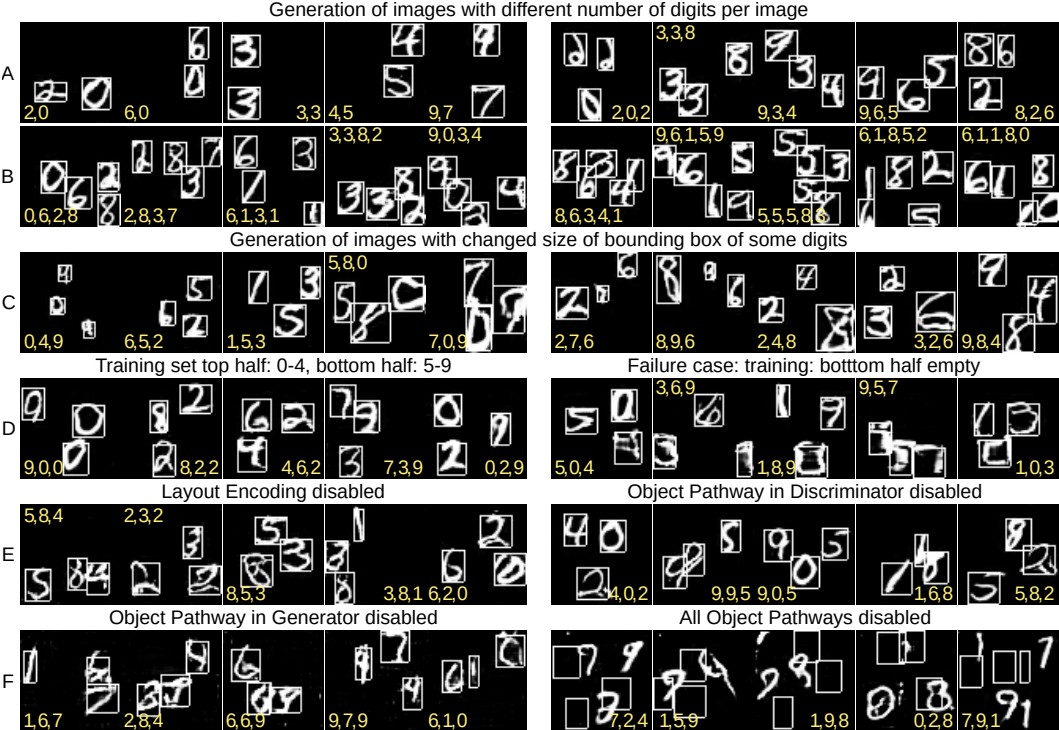

Figure 2: Multi-MNIST images generated by the model. Training included only images with three individual normal-sized digits. Highlighted bounding boxes and yellow ground truth for visualization.

As a first step, we tested whether our model can learn to generate digits at the specified locations and whether we can control the digit identity, the generated digit's size, and the number of generated digits per image. According to the results, we can control the location of individual digits, their identity, and their size, even though all training images contain exactly three digits in normal size. Figure 2 shows that we can control how many digits are generated within an image (rows A–B, for two to five digits) and various sizes of the bounding box (row C). As a second step, we created an additional Multi-MNIST data set in which all training images contain only digits 0–4 in the top half and only digits 5–9 in the bottom half of the image. For testing digits in the opposite half, we can see that the model is indeed capable of generalizing the position (row D, left), i.e. it can generate digits 0–4 in the bottom half of the image and digits 5–9 in the top half of the image. Nevertheless, we also observed that this does not always work perfectly, as the network sometimes alters digits towards the ones it has seen during training at the respective locations, e.g. producing a "4" more similar to a "9" if in bottom half of the image, or generating a "7" more similar to a "1" if in top half of the image.

As a next step, we created a Multi-MNIST data set with images that only contain digits in the top half of the image, while the bottom half is always empty. We can see (Figure 2, row D, right) that the resulting model is not able to generate digits in the bottom half of the image (see Figure 6 in the Appendix for more details on this). Controlling for the location still works, i.e. bounding boxes are filled with "something", but the digit identity is not clearly recognizable. Thus, the model is able to control both the object identity and the object location within an image and can generalize to novel object locations to some extent.

To test the impact of our model extensions, i.e. the object pathway in both the generator and the discriminator as well as the layout encoding, we performed ablation studies on the previously created Multi-MNIST data set with three digits at random locations. We first disabled the use of the layout encoding in the generator and left the rest of the model unchanged. In the results (Figure 2, row E, left), we can see that, overall, both the digit identity and the digit locations are still correct, but minor imperfections can be observed within various images. This is most likely due to the fact that the global pathway of the generator has no information about the digit identity and location until its features get merged with the object pathway. As a next test, we disabled the object pathway of the

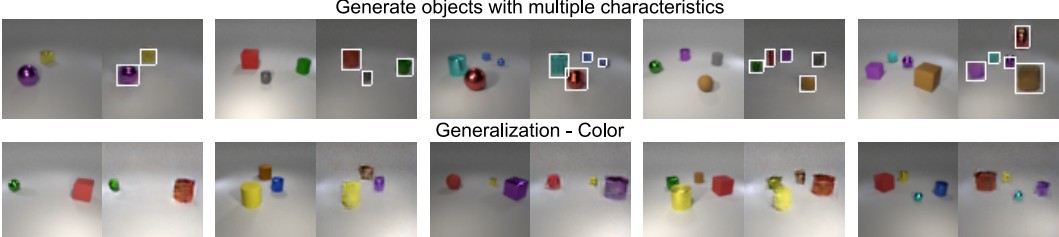

Figure 3: Images from the CLEVR data set. The left image of each pair shows the rendered image according to specific attributes. The right image of each pair is the image generated by our model.

discriminator and left the rest of the model unmodified. Again, we see (row E, right) that we can still control the digit location, although, again, minor imperfections are visible. More strikingly, we have a noticeably higher error rate in the digit identity, i.e. the wrong digit is generated at a given location, most likely due to the fact that there is not object pathway in the discriminator controlling the object identity at the various locations. In comparison, the imperfections are different when only the object pathway of the generator is disabled (row F, left). The layout encoding and the feedback of the discriminator seem to be enough to still produce the digits in the correct image location, but the digit identity is often incorrect or not recognizable at all. Finally, we tested disabling the object pathway in both the discriminator and the generator (see row F, right). This leads to a loss of control of both image location as well as identity and sometimes even results in images with more or fewer than three digits per image. This shows that only the layout encoding, without any of the object pathways, is not enough to control the digit identity and location. Overall, these results indicate that we do indeed need both the layout encoding, for a better integration of the global and object pathways, and the object pathways in both the discriminator and the generator, for optimal results.

## 4.2 CLEVR

In our second experiment we used more complex images containing multiple objects of different colors and shapes. The goal of this experiment was to evaluate the generalization ability of our object pathway across different object characteristics. For this, we performed tests similar to (Raj et al., 2017), albeit on the more complex CLEVR data set (Johnson et al., 2017). In the CLEVR data set objects are characterized by multiple properties, in our case the shape, the color, and the size. Based on the implementation provided by Johnson et al. (2017), we rendered 25,000 images with a resolution of $64 \times 64$ pixels containing $2 - 4$ objects per image. The label for a given bounding box of an object is the object shape and color (both encoded as one-hot encoding and then concatenated), while the object size is specified through the height and width of the bounding box.

Similar to the first experiment, we tested our model for controlling the object characteristics, size, and location. In the first row of Figure 3 we present the results of the trained model, where the left image of each pair shows the originally rendered one, while the right image was generated by our model. We can confirm that the model can control both the location and the objects' shape and color characteristics. The model can also generate images containing an arbitrary number of objects (forth and fifths pair), even though a maximum of four objects per image was seen during training.

The CLEVR data set offers a split specifically intended to test the generalization capability of a model, in which cylinders can be either red, green, purple, or cyan and cubes can be either gray, blue, brown, or yellow during training, while spheres can have any of these colors. During testing, the colors between cylinders and cubes are reversed. Based on these restrictions, we created a second data set of 25,000 training images for testing our model. Results of the test are shown in the second row of Figure 3 (again, left image of each pair shows the originally rendered one, while the right image was generated by our model). We can see that the color transfer to novel shape-color combinations takes place, but, similarly to the Multi-MNIST results, we can see some artifacts, where e.g. some cubes look a bit more like cylinders and vice versa. Overall, the CLEVR experiment confirms the indication that our model can control object characteristics (provided through labels) and object locations (provided through bounding boxes) and can generalize to novel object locations, novel amounts of objects per image, and novel object characteristic combinations within reasonable boundaries.

| Model | Resolution | IS ↑ | FID ↓ |
|---|---|---|---|
| GAN-INT-CLS Reed et al. (2016b) | $64 \times 64$ | $7.88 \pm 0.07$ | $60.62$ |
| StackGAN-V2 Zhang et al. (2018a) | $256 \times 256$ | $8.30 \pm 0.10$ | $81.59$ |
| StackGAN Zhang et al. (2018a) | $256 \times 256$ | $8.45 \pm 0.03$[1] | $74.05$ |
| PPGN Nguyen et al. (2017) | $227 \times 227$ | $9.58 \pm 0.21$ | |
| ChatPainter (StackGAN) Sharma et al. (2018) | $256 \times 256$ | $9.74 \pm 0.02$ | |
| Semantic Layout Hong et al. (2018b) | $128 \times 128$ | $11.46 \pm 0.09$[2] | |
| HDGan Zhang et al. (2018c) | $256 \times 256$ | $11.86 \pm 0.18$ | $71.27 \pm 0.12$[3] |
| AttnGAN Xu et al. (2018b) | $256 \times 256$ | $23.61 \pm 0.21$[4] | $33.10 \pm 0.11$[3] |
| *StackGAN + Object Pathways (Ours)*[5] | $256 \times 256$ | $12.12 \pm 0.31$ | $55.30 \pm 1.78$ |
| *AttnGAN + Object Pathways (Ours)* | $256 \times 256$ | $24.76 \pm 0.43$ | $33.35 \pm 1.15$ |

[1] Recently updated to $10.62 \pm 0.19$ in its source code.
[2] When using the ground truth bounding boxes at test time (as we do) the IS increases to $11.94 \pm 0.09$.
[3] FID score was calculated with samples generated with the pretrained model provided by the authors.
[4] The authors report a "best" value of $25.89 \pm 0.47$, but when calculating the IS with the pretrained model provided by the authors we only obtain an IS of $23.61$. Other researchers on the authors' Github website report a similar value for the pretrained model.
[5] We use the updated source code (IS of $10.62$) as our baseline model.

Table 1: Comparison of the Inception Score (IS) and Fréchet Inception Distance (FID) on the MS-COCO data set for different models. Note: the IS and FID values of our models are not necessarily directly comparable to the other models, since our model gets at test time, in addition to the image caption, up to three bounding boxes and their respective object labels as input.

## 4.3 MS-COCO

For our final experiment, we used the MS-COCO data set (Lin et al., 2014) to evaluate our model on natural images of complex scenes. In order to keep our evaluation comparable to previous work, we used the 2014 train/test split consisting of roughly 80,000 training and 40,000 test images and rescaled the images to a resolution of $256 \times 256$ px. At train-time, we used the bounding boxes and object labels of the three largest objects within an image, i.e. we used zero to three bounding boxes per image. Similarly to work by Johnson et al. (2018) we only considered objects that cover at least 2% of the image for the bounding boxes. To evaluate our results quantitatively, we computed both the Inception Score (IS, larger is better), which tries to evaluate how recognizable and diverse objects within images are (Salimans et al., 2016), as well as the Fréchet Inception Distance (FID, smaller is better), which compares the statistics of generated images with real images (Heusel et al., 2017). As a qualitative evaluation, we generated images that contain more than one object, and checked, whether the bounding boxes can control the object placement. We tested our approach with two commonly used architectures for text-to-image synthesis, namely the StackGAN (Zhang et al., 2017) and the AttnGAN (Xu et al., 2018b), and compared the images generated by these and our models.

In the StackGAN, the training process is divided into two steps: first, it learns a generator for images with a resolution of $64 \times 64$ px based on the image captions, and second, it trains a second generator, which uses the smaller images ($64 \times 64$ px) from the first generator and the image caption as input to generate images with a resolution of $256 \times 256$ px. Here, we added the object pathways and the layout encoding at the beginning of both the first generator and the second generator and used the object pathway in both discriminators. The other parts of StackGAN architecture and all hyperparameters remain the same as in the original training procedure for the MS-COCO data set. We trained the model three times from scratch and randomly sampled 3 times 30,000 image captions from the test set for each model. We then calculated the IS and FID values on each of the nine samples of 30,000 generated images and report the averaged values. As presented in Table 1, our StackGAN with added object pathways outperforms the original StackGAN both on the IS and the FID, increasing the IS from $10.62$ to $12.12$ and decreasing the FID from $74.05$ to $55.30$. Note, however, that this might also be due to the additional information our model is provided with as it receives up to three bounding boxes and respective bounding box labels per image in addition to the image caption.

We also extended the AttnGAN by Xu et al. (2018b), the current state-of-the-art model on the MS-COCO data set (based on the Inception Score), with our object pathway to evaluate its impact on

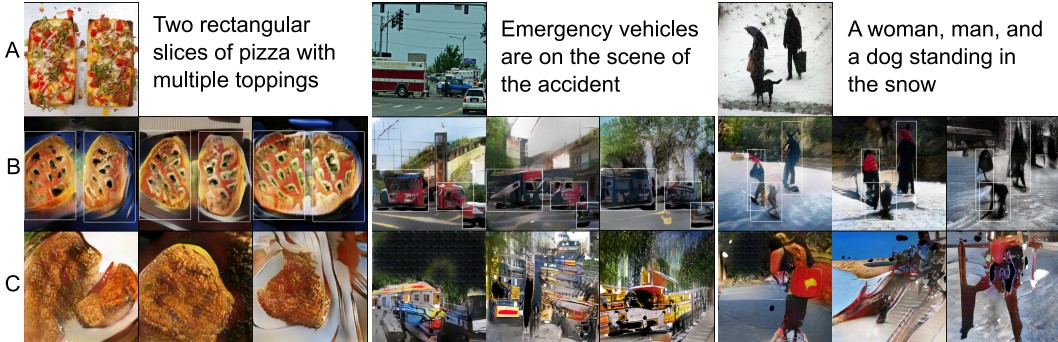

Figure 4: Examples of images generated from the given caption from the MS-COCO data set. *A)* shows the original images and the respective image captions, *B)* shows images generated by our StackGAN+OP (with the corresponding bounding boxes for visualization), and *C)* shows images generated by the original StackGAN (Zhang et al., 2017)[3]

a different model. As opposed to the StackGAN, the AttnGAN consists of only one model which is trained end-to-end on the image captions by making use of multiple, intermediate, discriminators. Three discriminators judge the output of the generator at an image resolution of $64 \times 64$, $128 \times 128$, and $256 \times 256$ px. Through this, the image generation process is guided at multiple levels, which helps during the training process. Additionally, the AttnGAN implements an attention technique through which the networks focus on specific areas of the image for specific words in the image caption and adds an additional loss that checks if the image depicts the content as described by the image caption. There, in the same way as for the StackGAN, we added our object pathway at the beginning of the generator as well as to the discriminator that judges the generator outputs at a resolution of $64 \times 64$ px. All other discriminators, the higher layers of the generator, and all other hyperparameters and training details stay unchanged. Table 1 shows that adding the object pathway to the AttnGAN increases the IS of our baseline model (the pretrained model provided by the authors) from $23.61$ to $24.76$, while the FID is roughly the same as for the baseline model.

To evaluate whether the StackGAN model equipped with an object pathway (StackGAN+OP) actually generates objects at the given positions we generated images that contain multiple objects and inspected them visually. Figure 4 shows some example images, more results can be seen in the Appendix in Figures 7 and 9. We can observe that the StackGAN+OP indeed generates images in which the objects are at appropriate locations. In order to more closely inspect our global and object pathways, we can also disable them during the image generation process. Figure 5 shows additional examples, in which we generate the same image with either the global or the object pathway disabled during the generation process. Row C of Figure 5 shows images in which the object pathway was disabled and, indeed, we observe that the images contain mostly background information and objects at the location of the bounding boxes are either not present or of much less detail than when the object pathway is enabled. Conversely, row D of Figure 5 shows images which were generated when the global pathway was disabled. As expected, areas outside of the bounding boxes are empty, but we also observe that the bounding boxes indeed contain images that resemble the appropriate objects. These results indicate, as in the previous experiments, that the global pathway does indeed model holistic image features, while the object pathway focuses on specific, individual objects.

When we add the object pathway to the AttnGAN (AttnGAN + OP) we can observe similar results[4]. Again, we are able to control the location and identity of objects through the object pathway, however, we observe that the AttnGAN+OP, as well as the AttnGAN in general, tends to place objects corresponding to specific features at many locations throughout the image. For example, if the caption contains the word "traffic light" the AttnGAN tends to place objects similar to traffic lights throughout the whole image. Since our model only focuses on generating objects at given locations, while not enforcing that these objects *only* occur at these locations, this behavior leads to the result that the AttnGAN+OP generates desired objects at the desired locations, but might also place the same object at other locations within the image. Note, however, that we only added the object pathway

---

[3]Generated with the model from: `https://github.com/hanzhanggit/StackGAN-Pytorch`

[4]Examples of images generated by the AttnGAN+OP can be seen in the Appendix in Figures 8 and 10.

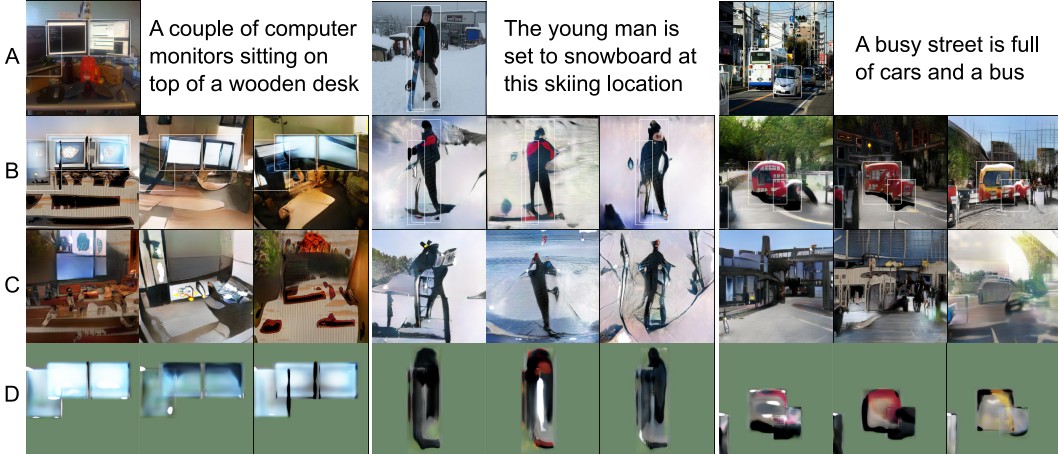

Figure 5: Examples of images generated from the given caption from the MS-COCO data set. *A)* shows the original images and the respective image captions, *B)* shows images generated by our StackGAN+OP (with the corresponding bounding boxes for visualization) with the object pathway enabled, *C)* shows images generated by the our StackGAN+OP when the object pathway is disabled, and *D)* shows images generated by the our StackGAN+OP when the global pathway is disabled.

to the lowest generator and discriminator and that we might gain even more control over the object location by introducing object pathways to the higher generators and discriminators, too.

In order to further evaluate the quality of the generations, we ran an object detection test on the generated images using a pretrained YOLOv3 network (Redmon & Farhadi, 2018). Here, the goal is to measure how often an object detection framework, which was trained on MS-COSO as well, can detect a specified object at a specified location[5]. The results confirm the previously made observations: For both the StackGAN and the AttnGAN the object pathway seems to improve the image quality, since YOLOv3 detects a given object more often correctly when the images are generated with an object pathway as opposed to images generated with the baseline models. The StackGAN generates objects at the given bounding box, resulting in an Intersection over Union (IoU) of greater than $0.3$ for all tested labels and greater than $0.5$ for $86.7\%$ of the tested labels. In contrast, the AttnGAN tends to place salient object features throughout the image, which leads to an even higher detection rate by the YOLOv3 network, but a smaller average IoU (only $53.3\%$ of the labels achieve an IoU greater than $0.3$). Overall, our experiments on the MS-COCO data set indicate that it is possible to add our object pathway to pre-existing GAN models without having to change the overall model architecture or training process. Adding the object pathway provides us with more control over the image generation process and can, in some cases, increase the quality of the generated images as measured via the IS or FID.

## 4.4 DISCUSSION

Our experiments indicate that we do indeed get additional control over the image generation process through the introduction of object pathways in GANs. This enables us to control the identity and location of multiple objects within a given image based on bounding boxes and thereby facilitates the generation of more complex scenes. We further find that the division of work on a global and object pathway seems to improve the image quality both subjectively and based on quantitative metrics such as the Inception Score and the Fréchet Inception Distance.

The results further indicate that the focus on global image statistics by the global pathway and the more fine-grained attention to detail of specific objects by the object pathway works well. This is visualized for example in rows C and D of Figure 5. The global pathway (row C) generates features for the general image layout and background but does not provide sufficient details for individual objects. The object pathway (row D), on the other hand, focuses entirely on the individual objects and generates features specifically for a given object at a given location. While this is the desired behavior

---

[5]See Appendix for more details on the procedure and the exact results.

of our model it can also lead to sub-optimal images if there are not bounding boxes for objects that should be present within the image. This can often be the case if the foreground object is too small (in our case less than 2% of the total image) and is therefore not specifically labeled. In this case, the objects are sometimes not modeled in the image at all, despite being prominent in the respective image caption, since the object pathway does not generate any features. We can observe this, for example, in images described as "*many sheep are standing on the grass*", where the individual sheep are too small to warrant a bounding box. In this case, our model will often only generate an image depicting grass and other background details, while not containing any sheep at all.

Another weakness is that bounding boxes that overlap too much (empirically an overlap of more than roughly 30%) also often lead to sub-optimal objects at that location. Especially in the overlapping section of bounding boxes we often observe local inconsistencies or failures. This might be the result of our merging of the different features within the object pathway since they are simply added to each other at overlapping areas. A more sophisticated merging procedure could potentially alleviate this problem.Another approach would be to additionally enhance the bounding box layout by predicting the specific object shape within each bounding box, as done for example by Hong et al. (2018b).

Finally, currently our model does not generate the bounding boxes and labels automatically. Instead, they have to be provided at test time which somewhat limits the usability for unsupervised image generation. However, even when using ground truth bounding boxes, our models still outperform other current approaches that are tested with ground truth bounding boxes (e.g. Hong et al. (2018b)) based on the IS and FID. This is even without the additional need of learning to specify the shape within each bounding box as done by Hong et al. (2018b). In the future, this limitation can be avoided by extracting the relevant bounding boxes and labels directly from the image caption, as it is done for example by Hong et al. (2018b), Xu et al. (2018a), and Tan et al. (2018).

## 5    CONCLUSION

With the goal of understanding how to gain more control over the image generation process in GANs, we introduced the concept of an additional object pathway. Such a mechanism for differentiating between a scene representation and object representations allows us to control the identity, location, and size of arbitrarily many objects within an image, as long as the objects do not overlap too strongly. In parallel, a global pathway, similar to a standard GAN, focuses on the general scene layout and generates holistic image features. The object pathway, on the other hand, gets as input an object label and uses this to generate features specifically for this object which are then placed at the location given by a bounding box The object pathway is applied iteratively for each object at each given location and as such, we obtain a representation of individual objects at individual locations and of the general image layout (background, etc.) as a whole. The features generated by the object and global pathway are then concatenated and are used to generate the final image output. Our tests on synthetic and real-world data sets suggest that the object pathway is an extension that can be added to common GAN architectures without much change to the original architecture and can, along with more fine-grained control over the image layout, also lead to better image quality.

### ACKNOWLEDGMENTS

The authors gratefully acknowledge partial support from the German Research Foundation DFG under project CML (TRR 169) and the European Union under project SECURE (No 642667). We also thank the NVIDIA Corporation for their support through the GPU Grant Program.

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

# A IMPLEMENTATION DETAILS

Here we provide some more details about the exact implementation of our experiments.

## A.1 MULTI-MNIST AND CLEVR

To train our GAN approach on the Multi-MNIST (CLEVR) data set we use the Stage-I Generator and Discriminator from the StackGAN MS-COCO architecture[6]. In our following description an **upsample block** describes the following sequence: nearest neighbor upsampling with factor 2, a convolutional layer with $X$ filters (filter size $3 \times 3$, stride 1, padding 1), batch normalization, and a ReLU activation. The bounding box labels are one-hot vectors of size $[1, 10]$ encoding the digit identity (CLEVR: $[1, 13]$ encoding object shape and color). Please refer to Table 2 for detailed information on the individual layers described in the following. For all leaky ReLU activations $alpha$ was set to $0.2$.

In the **object pathway of the generator** we first create a zero tensor $\mathbb{O}_G$ which will contain the feature representations of the individual objects. We then spatially replicate each bounding box label into a $4 \times 4$ layout of shape $(10, 4, 4)$ (CLEVR: $(13, 4, 4)$) and apply two upsampling blocks. The resulting tensor is then added to the tensor $\mathbb{O}_G$ at the location of the bounding box using a spatial transformer network.

In the **global pathway of the generator** we first obtain the layout encoding. For this we create a tensor of shape $(10, 16, 16)$ (CLEVR: $(13, 16, 16)$) that contains the one-hot labels at the location of the bounding boxes and is zero everywhere else. We then apply three convolutional layers, each followed by batch normalization and a leaky ReLU activation. We reshape the output to shape $(1, 64)$ and concatenate it with the noise tensor of shape $(1, 100)$ (sampled from a random normal distribution) to form a tensor of shape $(1, 164)$. This tensor is then fed into a dense layer, followed by batch normalization and a ReLU activation and the output is reshaped to $(-1, 4, 4)$. We then apply two upsampling blocks to obtain a tensor of shape $(-1, 16, 16)$.

At this point, the outputs of the object and the global pathway are concatenated along the channel axis to form a tensor of shape $(-1, 16, 16)$. We then apply another two upsampling blocks resulting in a tensor of shape $(-1, 64, 64)$ followed by a convolutional layer and a TanH activation to obtain the final image of shape $(-1, 64, 64)$.

In the **object pathway of the discriminator** we first create a zero tensor $\mathbb{O}_D$ which will contain the feature representations of the individual objects. We then use a spatial transformer network to extract the image features at the locations of the bounding boxes and reshape them to a tensor of shape $(1, 16, 16)$ (CLEVR: $(3, 16, 16)$). The one-hot label of each bounding box are spatially replicated to a shape of $(10, 16, 16)$ (CLEVR: $(13, 16, 16)$) and concatenated with the previously extracted features to form a tensor of shape $(11, 16, 16)$ (CLEVR: $(16, 16, 16)$). We then apply a convolutional layer, batch normalization and a leaky ReLU activation to the concatenation of features and label and, again, use a spatial transformer network to resize the output to the shape of the respective bounding box before adding it to the tensor $\mathbb{O}_D$.

In the **global pathway of the discriminator**, we apply two convolutional layers, each followed by batch normalization and a leaky ReLU activation and concatenate the resulting tensor with the output of the object pathway. After this, we again apply two convolutional layers, each followed by batch normalization and a leaky ReLU activation. We concatenate the resulting tensor with the conditioning information about the image content, in this case, the sum of all one-hot vectors. To this tensor we apply another convolutional layer, batch normalization, a leaky ReLU activation, and another convolutional layer, to obtain the final output of the discriminator of shape $(1)$.

Similarly to the procedure of StackGAN and other conditional GANs we train the discriminator to classify real images with correct labels (the sum of one-hot vectors supplied in the last step of the process) as real, while generated images with correct labels and real images with (randomly sampled) incorrect labels should be classified as fake.

---

[6] https://github.com/hanzhanggit/StackGAN-Pytorch

## A.2 MS-COCO

**StackGAN-Stage-I**   For training the Stage-I generator and discriminator (images of size $64 \times 64$ pixels) we follow the same procedure and architecture outlined in the previous section about the training on the Multi-MNIST and CLEVR data sets. The only difference is that we now have **image captions** as an additional description of the image. As such, to obtain the bounding box labels we concatenate the image caption embedding[7] and the one-hot encoded bounding box label and apply a dense layer with $128$ units, batch normalization, and a ReLU activation to it, to obtain a label of shape $(1, 128)$ for each bounding box. In the final step of the discriminator when we concatenate the feature representation with the conditioning vector, we use the image encoding as conditioning vector and do not use any bounding box labels at this step. The rest of the training proceeds as described in the previous section, except that the bounding box labels now have a shape of $(1, 128)$. All other details can be found in Table 2.

**StackGAN-Stage-II**   In the second part of the training, we train a second generator and discriminator to generate images with a resolution of $256 \times 256$ pixels. The generator gets as input images with a resolution of $64 \times 64$ pixels (generated by the trained Stage-I generator) and the image caption and uses them to generate images with a $256 \times 256$ pixels resolution. A new discriminator is trained to distinguish between real and generated images.

On the **Stage-II generator** we perform the following modifications we use the same procedure as in the Stage-I generator to obtain the **bounding box labels**. To obtain an **image encoding** from the generated $64 \times 64$ image we use three convolutional layers, each followed by batch normalization and a ReLU activation to obtain a feature representation of shape $[-1, 16, 16]$. Additionally, we replicate each bounding box label (obtained with the dense layer) spatially at the locations of the bounding boxes on an empty canvas of shape $[128, 16, 16]$ and then concatenate it along the channel axis with the image encoding and the spatially replicated image caption embedding. As in the standard StackGAN we then apply more convolutional layers with residual connections to obtain the final image embedding of shape $[-1, 16, 16]$, which provides the input for both the object and the global pathway.

The **generator's object pathway** gets as input the image encoding described in the previous step. First, we create a zero tensor $\mathbb{O}_G$ which will contain the feature representations of the individual objects. We then use a spatial transformer network to extract the features from within the bounding box and reshapes those features to $[-1, 16, 16]$. After this, we apply two upsample blocks and then use a spatial transformer network to add the features to $\mathbb{O}_G$ within the bounding box region. This is done for each of the bounding boxes within the image.

The **generator's global pathway** gets as input the image encoding and uses the same convolutional layers and upsampling procedures as the original StackGAN Stage-II generator. The **outputs of the object and global pathway** are merged at the resolution of $[-1, 64, 64]$ by concatenating the two outputs along the channel axis. After this, we continue using the standard StackGAN architecture to generate images of shape $[3, 256, 256]$.

The **Stage-II discriminator's object pathway** first creates a zero tensor $\mathbb{O}_D$ which will contain the feature representations of the individual objects. It gets as input the image (resolution of $256 \times 256$ pixels) and we use a spatial transformer network to extract the features from the bounding box and reshape those features to a shape of $[3, 32, 32]$. We spatially replicate the bounding box label (one-hot encoding) to a shape of $[-1, 32, 32]$ and concatenate it with the extracted features along the channel axis. This is then given to the object pathway which consists of two convolutional layers with batch normalization and a LeakyReLU activation. The output of the object pathway is again transformed to the width and height of the bounding box with a spatial transformer network and then added to $\mathbb{O}_D$. This procedure is performed with each of the bounding boxes within the image (maximum of three during training).

The **Stage-II discriminator's global pathway** consists of the standard StackGAN layers, i.e. it gets as input the image ($256 \times 256$ pixels) and applies convolutional layers with stride 2 to it. The **outputs of the object and global pathways** are merged at the resolution of $[-1, 32, 32]$ by concatenating the

---

[7]Downloaded from `https://github.com/reedscot/icml2016`

two outputs along the channel axis We then apply more convolutional with stride 2 to decrease the resolution. After this, we continue in the same way as the original StackGAN.

**AttnGAN** On the AttnGAN[8] we only modify the training at the lower layers of the generator and the first discriminator (working on images of $64 \times 64$ pixels resolution). For this, we perform the same modifications as described in the StackGAN-Stage-I generator and discriminator. In the **generator** we obtain the bounding box labels in the same way as in the StackGAN, by concatenating the image caption embedding with the respective one-hot vector and applying a dense layer with 100 units, batch normalization, and a ReLU activation to obtain a bounding box label. In contrast to the previous architectures, we follow the AttnGAN implementation in use the gated linear unit function (GLU) as standard activation for our convolutional layers in the generator.

In the **generator's object pathway** we first create a zero tensor $\mathbb{O}_G$ of shape $(192, 16, 16)$ which will contain the feature representations of the individual objects. We then spatially replicate each bounding box label into a $4 \times 4$ layout of shape $(100, 4, 4)$ and apply two upsampling blocks with 768 and 384 filters (filter size=3, stride=1, padding=1). The resulting tensor is then added to the tensor $\mathbb{O}_G$ at the location of the bounding box using a spatial transformer network.

In the **global pathway of the generator** we first obtain the layout encoding in the same way as in the StackGAN-I generator, except that the three convolutional layers of the layout encoding now have 50, 25, and 12 filters respectively (filter size=3, stride=2, padding=1). We concatenate it with the noise tensor of shape $(1, 100)$ (sampled from a random normal distribution) and the image caption embedding to form a tensor of shape $(1, 248)$. This tensor is then fed into a dense layer with 24,576 units, followed by batch normalization and a ReLU activation and the output is reshaped to $(768, 4, 4)$. We then apply two upsampling blocks with 768 and 384 filters to obtain a tensor of shape $(192, 16, 16)$.

At this point the **outputs of the object and the global pathways** are concatenated along the channel axis to form a tensor of shape $(384, 16, 16)$. We then apply another two upsampling blocks with 192 and 96 filters, resulting in a tensor of shape $(48, 64, 64)$. This feature representation is then used by the following layers of the AttnGAN generator in the same way as detailed in the original paper and implementation.

In the **object pathway of the discriminator** we first create a zero tensor $\mathbb{O}_D$ which will contain the feature representations of the individual objects. We then use a spatial transfomer network to extract the image features at the locations of the bounding boxes and reshape them to a tensor of shape $(3, 16, 16)$. The one-hot label of each bounding box is spatially replicated to a shape of $(-1, 16, 16)$ and concatenated with the previously extracted features. We then apply a convolutional layer with 192 filters (filter size=4, stride=1, padding=1), batch normalization and a leaky ReLU activation to the concatenation of features and label and, again, use a spatial transformer network to resize the output to the shape of the respective bounding box before adding it to the tensor $\mathbb{O}_D$.

In the **global pathway of the discriminator** we apply two convolutional layers with 96 and 192 filters (filter size=4, stride=2, padding=1), each followed by batch normalization and a leaky ReLU activation and concatenate the resulting tensor with the output of the object pathway. After this, we again apply two convolutional layers with 384 and 768 filters (filter size=4, stride=2, padding=1), each followed by batch normalization and a leaky ReLU activation. We concatenate the resulting tensor with the spatially replicated image caption embedding. To this tensor we apply another convolutional layer with 768 filters (filter size=3, stride=1, padding=1), batch normalization, a leaky ReLU activation, and another convolutional layer with one filter (filter size=4, stride=4, padding=0), to obtain the final output of the discriminator of shape $(1)$. The rest of the training and all other hyperparameters and architectural values are left the same as in the original implementation.

---

[8]`https://github.com/taoxugit/AttnGAN`

| | Multi-MNIST | CLEVR | MS-COCO-I | MS-COCO-II |
|---|---|---|---|---|
| Optimizer | Adam ($beta_1 = 0.5$, $beta_2 = 0.999$) | | | |
| Learning Rate | 0.0002 | 0.0002 | 0.0002 | 0.0002 |
| Schedule: halve every $x$ epochs | 10 | 20 | 20 | 20 |
| Training Epochs | 20 | 40 | 120 | 110 |
| Batch Size | 128 | 128 | 128 | 40 |
| Weight Initialization | $\mathcal{N}(0, 0.02)$ | $\mathcal{N}(0, 0.02)$ | $\mathcal{N}(0, 0.02)$ | $\mathcal{N}(0, 0.02)$ |
| Z-Dim / Img-Caption-Dim | 100 / 10 | 100 / 13 | 100 / 128 | 100 / 128 |
| **Generator** | | | | |
| Image Encoder | | | | |
| Conv (fs=3, s=1, p=1) | | | | 192 |
| Conv (fs=4, s=2, p=1) | | | | 384 |
| Conv (fs=4, s=2, p=1) | | | | 768 |
| Concat with image caption and bbox labels | | | | $(1024, 16, 16)$ |
| Conv (fs=3, str=1, pad=1) | | | | 768 |
| $4 \times$ Res. (fs=3, s=1, p=1) | | | | 768 |
| Object Pathway | | | | |
| $\mathbb{O}_G$ Shape | $(256, 16, 16)$ | $(192, 16, 16)$ | $(384, 16, 16)$ | $(192, 64, 64)$ |
| Upsample (fs=3, s=1, p=1) | 512 | 384 | 768 | 384 |
| Upsample (fs=3, s=1, p=1) | 256 | 192 | 384 | 192 |
| Output Shape | $(256, 16, 16)$ | $(192, 16, 16)$ | $(384, 16, 16)$ | $(192, 64, 64)$ |
| Global Pathway | | | | |
| Layout Encoding | | | | |
| Conv (fs=3, s=2, p=1) | 64 | 64 | 64 | |
| Conv (fs=3, s=2, p=1) | 32 | 32 | 32 | |
| Conv (fs=3, s=2, p=1) | 16 | 16 | 16 | |
| Dense Layer Units | 16,384 | 12,288 | 24,576 | |
| Upsample (fs=3, s=1, p=1) | 512 | 384 | 768 | 384 |
| Upsample (fs=3, s=1, p=1) | 256 | 192 | 384 | 192 |
| Output Shape | $(256, 16, 16)$ | $192, 16, 16)$ | $(384, 16, 16)$ | $(192, 64, 64)$ |
| Concat outputs of object and global pathways | $(512, 16, 16)$ | $(384, 16, 16)$ | $(768, 16, 16)$ | $(384, 64, 64)$ |
| Upsample (fs=3, s=1, p=1) | 128 | 96 | 192 | 96 |
| Upsample (fs=3, s=1, p=1) | 64 | 48 | 96 | 48 |
| Conv (fs=3, s=1, p=1) | 1 | 3 | 3 | 3 |
| Generator Output | $(1, 64, 64)$ | $(3, 64, 64)$ | $(3, 64, 64)$ | $(3, 256, 256)$ |
| **Discriminator** | | | | |
| Object Pathway | | | | |
| $\mathbb{O}_D$ Shape | $(128, 16, 16)$ | $(96, 16, 16)$ | $(192, 16, 16)$ | $(192, 32, 32)$ |
| Conv (fs=4, s=1, p=1) | 128 | 96 | 192 | 192 |
| Conv (fs=4, s=1, p=1) | | | | 192 |
| Output Shape | $(128, 16, 16)$ | $(96, 16, 16)$ | $(192, 16, 16)$ | $(192, 32, 32)$ |
| Global Pathway | | | | |
| Conv (fs=4, s=2, p=1) | 64 | 48 | 96 | 96 |
| Conv (fs=4, s=2, p=1) | 128 | 96 | 192 | 192 |
| Conv (fs=4, s=2, p=1) | | | | 384 |
| Output Shape | $(128, 16, 16)$ | $(96, 16, 16)$ | $(192, 16, 16)$ | $(384, 32, 32)$ |
| Concat outputs of object and global pathways | $(256, 16, 16)$ | $(192, 16, 16)$ | $(384, 16, 16)$ | $(576, 32, 32)$ |
| Conv (fs=4, s=2, p=1) | 256 | 192 | 384 | 768 |
| Conv (fs=4, s=2, p=1) | 512 | 384 | 768 | 1,536 |
| Conv (fs=4, s=2, p=1) | | | | 3,072 |
| Conv (fs=3, s=1, p=1) | | | | 1,536 |
| Conv (fs=3, s=1, p=1) | | | | 768 |
| Concat with conditioning vector | $(522, 4, 4)$ | $(397, 4, 4)$ | $(896, 4, 4)$ | $(896, 4, 4)$ |
| Conv (fs=3, s=1, p=1) | 512 | 384 | 768 | 768 |
| Conv (fs=4, s=4, p=0) | 1 | 1 | 1 | 1 |

Table 2: Overview of the individual layers used in our networks to generate images of resolution $64 \times 64$ / $256 \times 256$ pixels. Values in brackets $(C, H, W)$ represent the tensor's shape. Numbers in the columns after convolutional, residual, or dense layers describe the number of filters / units in that layer. (fs=$x$, s=$y$, p=$z$) describes filter size, stride, and padding for that convolutional / residual layer.

## B    ADDITIONAL EXAMPLES OF MULTI-MNIST RESULTS: TRAINING AND TEST SET OVER COMPLEMENTARY REGIONS

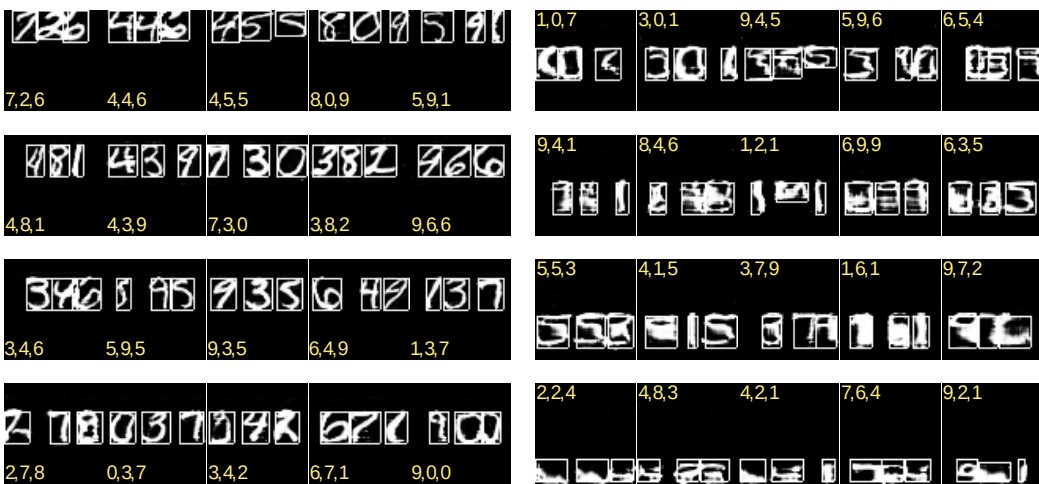

Figure 6: Systematic test of digits over vertically different regions. Training set included three normal-sized digits only in the top half of the image. Highlighted bounding boxes and yellow ground truth for visualization. We can see that the model fails to generate recognizable digits once their location is too far in the bottom half of the image, as this location was never observed during training.

## C    ADDITIONAL EXAMPLES OF MS-COCO RESULTS: STACKGAN

Figure 7 shows results of text-to-image synthesis on the MS-COCO data set with the StackGAN architecture. Rows A show the original image and image caption, rows B show the images generated by our StackGAN + Object Pathway and the given bounding boxes for visualization, and rows C show images generated by the original StackGAN (pretrained model obtained from `https://github.com/hanzhanggit/StackGAN-Pytorch`). The last block of examples (last row) show typical failure cases of our model, where there is no bounding box for the foreground object present. As a result our model only generates the background, without the appropriate foreground object, even though the foreground object is very clearly described in the image caption. Figure 9 provides similar results but for random bounding box positions. The first six examples show images generated by our StackGAN where we changed the location and size of the respective bounding boxes. The last three examples show failure cases in which we changed the location of the bounding boxes to "unusual" locations. For the image with the child on the bike, we put the bounding box of the bike somewhere in the top half of the image and the bounding box for the child somewhere in the bottom part. Similarly, for the man sitting on a bench, we put the bench in the top and the man in the bottom half of the image. Finally, for the image depicting a pizza on a plate, we put the plate location in the top half of the image and the pizza in the bottom half.

## D    ADDITIONAL EXAMPLES OF MS-COCO RESULTS: ATTNGAN

Figure 8 shows results of text-to-image synthesis on the MS-COCO data set with the AttnGAN architecture. Rows A show the original image and image caption, rows B show the images generated by our AttnGAN + Object Pathway and the given bounding boxes for visualization, and rows C show images generated by the original AttnGAN (pretrained model obtained from `https://github.com/taoxugit/AttnGAN`). The last block of examples (last row) show typical failure cases, in which the model does generate the appropriate object within the bounding box, but also places the same object at multiple other locations within the image. Similarly as for StackGAN, Figure 10 shows images generated by our AttnGAN where we randomly change the location of the various bounding boxes. Again, the last three examples show failure cases where we put the locations of the bounding boxes at "uncommon" positions. In the image depicting the sandwiches we put the location of the plate in the top half of the image, in the image with the dogs we put the dogs' location in the top half, and in the image with the motorbike we put the human in the left half and the motorbike in the right half of the image.

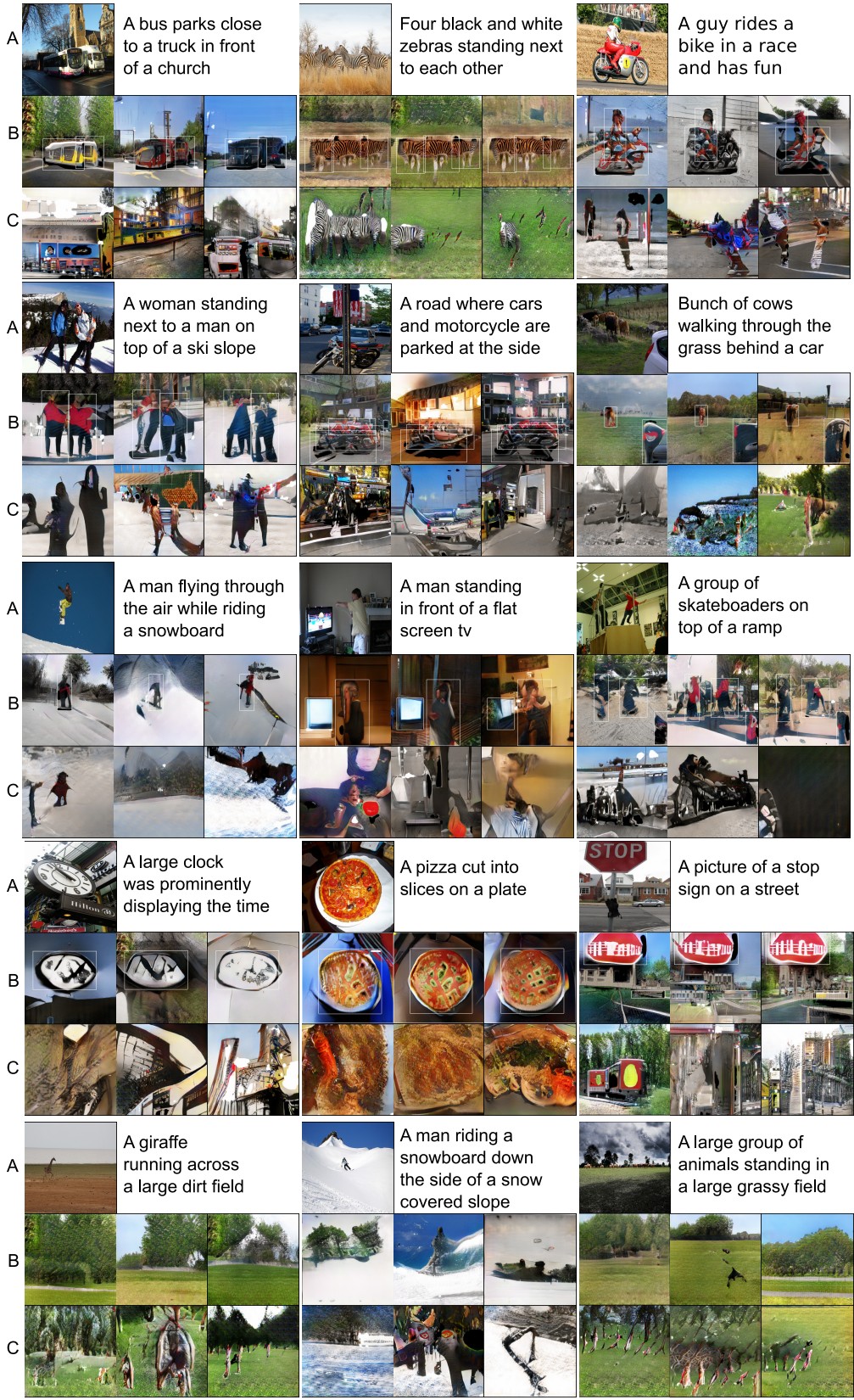

Figure 7: Additional StackGAN examples – refer to page 17 for information about the figure.

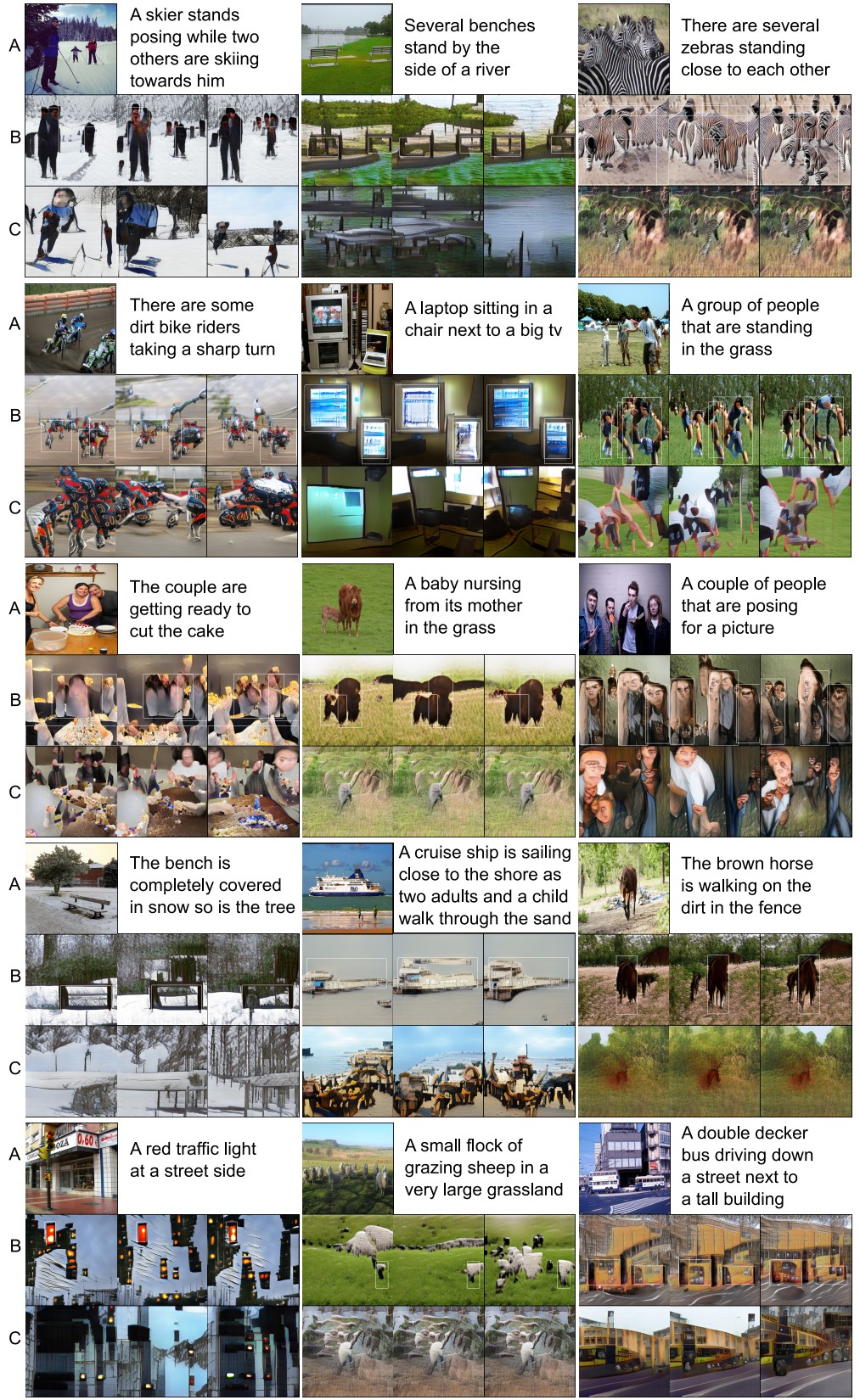

Figure 8: Additional AttnGAN examples – refer to page 17 for more information about the figure.

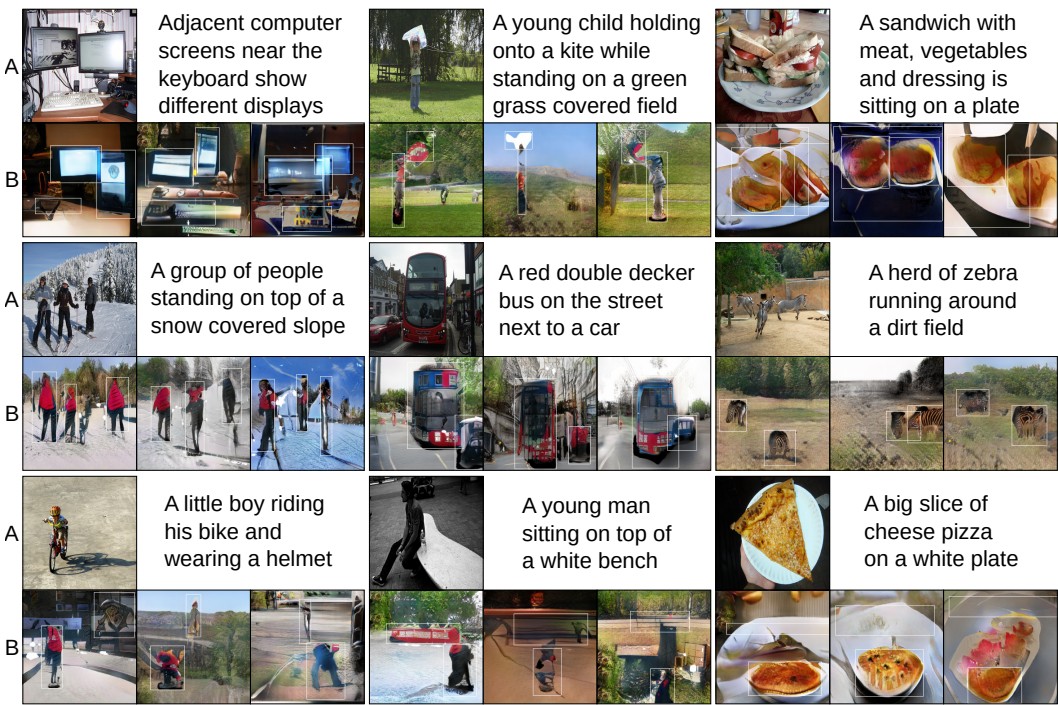

Figure 9: StackGAN examples with random locations – refer to page 17 for more information.

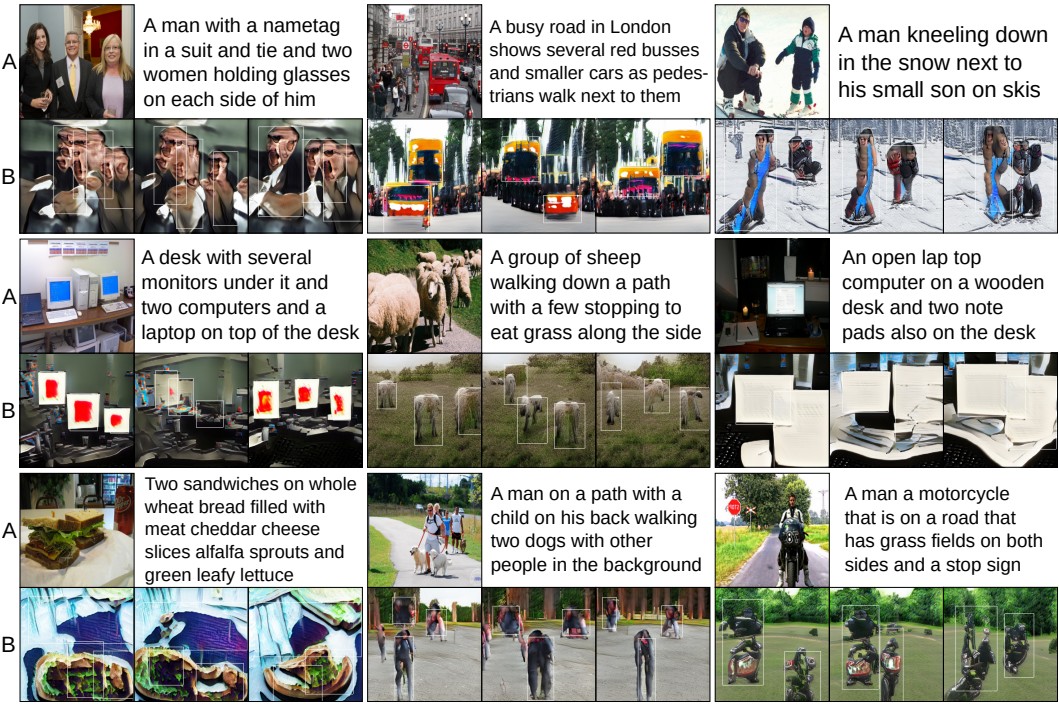

Figure 10: AttnGAN examples with random locations – refer to page 17 for more information.

| Label | Occurrences | Words in captions | Label | Occurrences | Words in captions |
|---|---|---|---|---|---|
| Person | 13773 | person, people, human, man, men, woman, women, child | Umbrella | 727 | umbrella |
| | | | Elephant | 708 | elephant |
| | | | Chair | 632 | chair, stool |
| Dining table | 3130 | table, desk | Zebra | 627 | zebra |
| Car | 1694 | car, auto, vehicle, cab | Boat | 627 | boat |
| Cat | 1658 | cat | Bird | 610 | bird |
| Dog | 1543 | dog | Aeroplane | 602 | plane |
| Bus | 1198 | bus | Bicycle | 600 | bicycle |
| Train | 1188 | train | Surfboard | 595 | surfboard |
| Bed | 984 | bed | Kite | 593 | kite |
| Pizza | 906 | pizza | Truck | 561 | truck |
| Horse | 874 | horse | Stop sign | 522 | stop |
| Giraffe | 828 | giraffe | TV Monitor | 471 | tv, monitor, screen |
| Toilet | 797 | toilet | Sofa | 467 | sofa, couch |
| Bear | 777 | bear | Sandwich | 387 | sandwich |
| Bench | 732 | bench | Sheep | 368 | sheep |

Table 3: Words that were used to identify given labels in the image caption for the YOLOv3 object detection test.

## E  OBJECT DETECTION ON MS-COCO IMAGES

To further inspect the quality of the location and recognizability of the generated objects within an image, we ran a test on object detection using a YOLOv3 network Redmon & Farhadi (2018) that was also pretrained on the MS-COCO data set[9]. We use the Pytorch implementation from `https://github.com/ayooshkathuria/pytorch-yolo-v3` to get the bounding box and label predictions for our images. We follow the standard guidelines and keep all hyperparameters for the YOLOv3 network as in the implementation. We picked the 30 most common training labels (based on how many captions contain these labels) and evaluate the models on these labels, see Table 3.

In the following, we evaluate how often the pretrained YOLOv3 network recognizes a specific object within a generated image that should contain this object based on the image caption. For example, we expect an image generated from the caption *"a young woman taking a picture with her phone"* to contain a person somewhere in the image and we check whether the YOLOv3 network actually recognizes a person in the generated image. Since the baseline StackGAN and AttnGAN only receive the image caption as input (no bounding boxes and no bounding box labels) we decided to only use captions that clearly imply the presence of the given label (see Table 3). We chose this strategy in order to allow for a fair comparison of the resulting presence or absence of a given object. Specifically, for a given label we choose all image captions from the test set that contain one of the associated words for this label (associated words were chosen manually, see Table 3) and then generated three images for each caption with each model. Finally, we counted the number of images in which the given object was detected by the YOLOv3 network. Table 4 shows the ratio of images for each label and each model in which the given object was detected at any location within the image.

Additionally, for our models that also receive the bounding boxes as input, we calculated the Intersection over Union (IoU) between the ground truth bounding box (the bounding box supplied to the model) and the bounding box predicted by the YOLOv3 network for the recognized object. Table 4 presents the average IoU (for the models that have an object pathway) for each object in the images in which YOLOv3 detected the given object. For each image in which YOLOv3 detected the given object, we calculated the IoU between the predicted bounding box and the ground truth bounding box for the given object. In the cases in which either an image contains multiple instances of the given object (i.e. multiple different bounding boxes for this object were given to the generator) or YOLOv3 detects the given object multiple times we used the maximum IoU between all predicted and ground truth bounding boxes for our statistics.

---

[9]Pretrained weights from the author, acquired via: `https://pjreddie.com/darknet/yolo/`

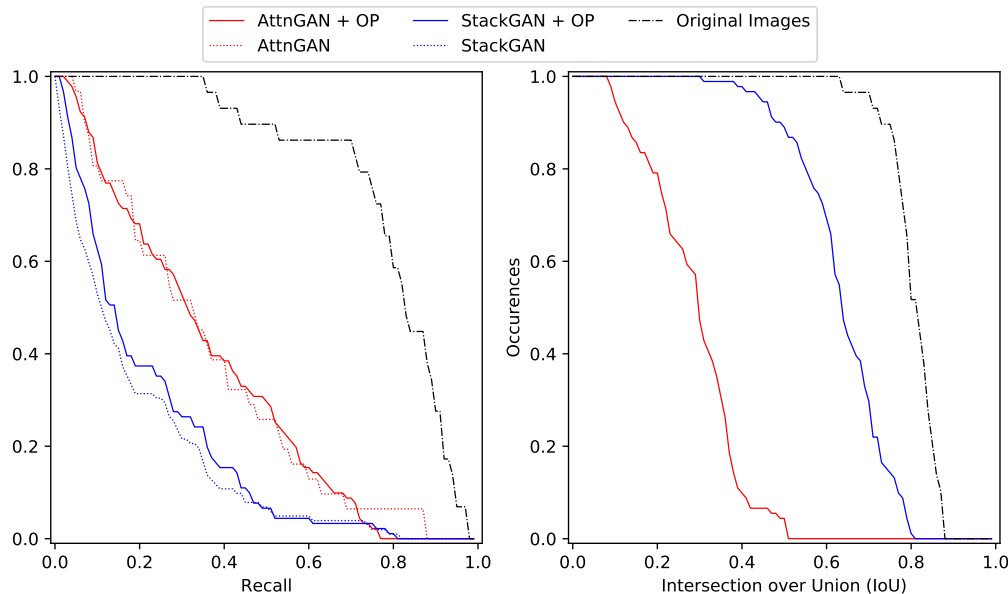

Figure 11: Distribution of recall and IoU values in the YOLOv3 object detection test.

Figure 11 visualizes how the IoU and recall values are distributed for the different models, and Table 4 summarizes the results with the 30 tested labels. We can observe that the StackGAN with object pathway outperforms the original StackGAN when comparing the recall of the YOLOv3 network, i.e. in how many images with a given label the YOLOv3 network actually detected the given object. The recall of the original StackGAN is higher than $10\%$ for $26.7\%$ of the labels, while our StackGAN with object pathway results in a recall greater than $10\%$ for $60\%$ of the labels. The IoU is greater than $0.3$ for every label, while $86.7\%$ of the labels result an IoU of greater than $0.5$ (original images: $100\%$) and $30\%$ have an IoU of greater than $0.7$ (original images: $96.7\%$). This indicates that we can indeed control the location and identity of various objects within the generated images.

Compared to the StackGAN, the AttnGAN achieves a much greater recall, with $80\%$ and $83.3\%$ of the labels having a recall of greater than $10\%$ for the original AttnGAN and the AttnGAN with object pathway respectively. The difference in recall values between the original AttnGAN and the AttnGAN with object pathway is also smaller, with our AttnGAN having a higher (lower) recall than the original AttnGAN (we only count cases where the difference is at least $5\%$) in $26.7\%$ ($13.3\%$) of the labels. The average IoU, on the other hand, is a lot smaller for the AttnGAN than for the StackGAN. We only achieve an IoU greater than $0.3$ ($0.5$, $0.7$) for $53.3\%$ ($3.3\%$, $0\%$) of the labels. As mentioned in the discussion (subsection 4.4), we attribute this to the observation that the AttnGAN tends to place seemingly recognizable features of salient objects at arbitrary locations throughout the image. This might attribute to the overall higher recall but may negatively affect the IoU.

Overall, these results further confirm our previous experiments and highlight that the addition of the object pathway to the different models does not only enable the direct control of object location and identity but can also help to increase the image quality. The increase in image quality is supported by a higher Inception Score, lower Fréchet Inception Distance (for StackGAN) and a higher performance of the YOLOv3 network in detecting objects within generated images.

| Label | Orig. Img. Recall | IoU | StackGAN Recall | StackGAN + OP Recall | StackGAN + OP IoU | AttnGAN Recall | AttnGAN + OP Recall | AttnGAN + OP IoU |
|---|---|---|---|---|---|---|---|---|
| Person | .943 | .824 | .355 | .451 ± .019 | .624 ± .012 | .598 | .610 ± .008 | .276 ± .006 |
| Dining table | .355 | .774 | .007 | .022 ± .004 | .734 ± .011 | .069 | .045 ± .022 | .490 ± .018 |
| Car | .433 | .792 | .012 | .047 ± .007 | .622 ± .020 | .006 | .063 ± .010 | .144 ± .043 |
| Cat | .715 | .821 | .021 | .104 ± .100 | .622 ± .008 | .423 | .430 ± .066 | .350 ± .012 |
| Dog | .703 | .819 | .068 | .150 ± .007 | .601 ± .004 | .450 | .488 ± .048 | .311 ± .007 |
| Bus | .747 | .877 | .161 | .393 ± .031 | .794 ± .009 | .352 | .416 ± .032 | .374 ± .006 |
| Train | .900 | .835 | .133 | .310 ± .033 | .700 ± .007 | .393 | .438 ± .110 | .355 ± .036 |
| Bed | .775 | .789 | .032 | .141 ± .018 | .701 ± .001 | .539 | .552 ± .030 | .505 ± .002 |
| Pizza | .912 | .842 | .119 | .485 ± .101 | .786 ± .004 | .444 | .660 ± .054 | .395 ± .016 |
| Horse | .933 | .842 | .129 | .330 ± .048 | .585 ± .039 | .532 | .619 ± .027 | .300 ± .006 |
| Giraffe | .972 | .857 | .173 | .467 ± .035 | .606 ± .030 | .472 | .650 ± .084 | .365 ± .030 |
| Toilet | .898 | .826 | .005 | .122 ± .021 | .690 ± .010 | .201 | .220 ± .021 | .224 ± .011 |
| Bear | .381 | .859 | .015 | .120 ± .018 | .720 ± .036 | .319 | .303 ± .028 | .357 ± .010 |
| Bench | .828 | .798 | .001 | .030 ± .008 | .627 ± .034 | .094 | .094 ± .031 | .308 ± .018 |
| Umbrella | .912 | .762 | .001 | .023 ± .009 | .578 ± .030 | .060 | .063 ± .017 | .154 ± .053 |
| Elephant | .940 | .867 | .060 | .414 ± .069 | .688 ± .033 | .350 | .500 ± .141 | .353 ± .006 |
| Chair | .757 | .755 | .014 | .039 ± .004 | .488 ± .039 | .070 | .093 ± .005 | .225 ± .001 |
| Zebra | .972 | .875 | .732 | .781 ± .023 | .686 ± .017 | .870 | .766 ± .063 | .315 ± .022 |
| Boat | .795 | .709 | .077 | .010 ± .011 | .594 ± .021 | .168 | .202 ± .027 | .206 ± .020 |
| Bird | .837 | .781 | .059 | .097 ± .027 | .500 ± .066 | .322 | .357 ± .042 | .250 ± .020 |
| Aeroplane | .912 | .812 | .125 | .223 ± .043 | .667 ± .026 | .499 | .415 ± .010 | .320 ± .035 |
| Bicycle | .825 | .760 | .007 | .053 ± .020 | .558 ± .052 | .170 | .191 ± .013 | .233 ± .024 |
| Surfboard | .873 | .780 | .030 | .067 ± .019 | .459 ± .056 | .104 | .110 ± .025 | .143 ± .016 |
| Kite | .772 | .633 | .029 | .057 ± .028 | .426 ± .086 | .260 | .162 ± .068 | .120 ± .018 |
| Truck | .887 | .832 | .082 | .243 ± .062 | .717 ± .022 | .378 | .367 ± .027 | .393 ± .019 |
| Stop Sign | .527 | .874 | .001 | .261 ± .057 | .780 ± .011 | .070 | .124 ± .048 | .101 ± .014 |
| TV Monitor | .818 | .833 | .037 | .264 ± .005 | .765 ± .016 | .529 | .435 ± .314 | .243 ± .066 |
| Sofa | .878 | .794 | .012 | .087 ± .024 | .628 ± .044 | .170 | .191 ± .057 | .329 ± .028 |
| Sandwich | .792 | .796 | .045 | .139 ± .049 | .628 ± .014 | .340 | .370 ± .054 | .318 ± .031 |
| Sheep | .943 | .727 | .004 | .091 ± .006 | .460 ± .011 | .250 | .304 ± .037 | .116 ± .022 |

Table 4: Results of YOLOv3 detections on generated and original images. Recall provides the fraction of images in which YOLOv3 detected the given object. *IoU* (Intersection over Union) measures the maximum IoU per image in which the given object was detected.

