# OpenReview forum: "Generating Multiple Objects at Spatially Distinct Locations"
_ICLR.cc/2019/Conference_

### Official Review · AnonReviewer1 · 2018-11-01
**Review for "Generating Multiple Objects at Spatially Distinct Locations"**

**Rating:** 7
**Confidence:** 4

**Review:**

The authors are proposing a method for allowing the generation of multiple objects in generated images given simple supervision such as bounding boxes and their associated labels. They control the spatial location of generated objects by the mean of an object pathway added to the architecture of both Generator and Discriminator within a GAN framework. They show generated results on Multi-MNIST, CLEVR with discussions of their model's abilities and properties. they also provide quantitative results on MSCOCO (IS and FID) using StackGAN and AttGAN models with the object pathway modifications and show some improvements compared to the original models. However it must be noted (as commented by the authors) that these models are using image captions only and do not have explicit supervision of bounding box and object labels.

This paper proposes a simple approach to generating requested objects in GAN-based image generation task, The method is supervised and requests (in its current form) the Bounding Boxes and Labels of the objects to integrate into the image generation. This task of controlling the nature (identity) and size of objects to integrate in a generated image is an important one and is significant to the GAN-base image generation community. In terms of originality, the approach is a nice simple architecture that takes care of the spatial location problem head-on. It seems like an obvious step but this does not take away from the merits of the proposed method.

The generator Global path is given a noise component. From the text, it does not seem that the Object path is given a noise component. Do you generate always the same object given the same label and Bounding Box then? Why not integrate some noise in this pipeline too?


Multi-MNIST:
The authors present results on Multi-MNIST 50K customed data to present the ability of the model to accurately put request images in the correct bounding box (BB) and do some ablation study. This is an interesting test as it shows that indeed the method proposed generates digits where it is expected to. Could you provide the ground truth labels for each/some image/s? For the failure cases it is often not clear what digit is what. For the Row E and F, 1s could be 7s and vice versa. Since it is a qualitative study, it would be nice to have the Ground Truth (GT) (which you provide to G at for generation). For the failure case of Row D (right) an interesting results would have been to have example of a digit bounding box from top to bottom with few pixel vertical shift to visualize when the model starts to mess-up the generation. This seems to point that your model (exposed to the location from BB for the object paths) is sensitive to what locations it has seen in training. How would you make the object path more robust to unseen location (overall you need to design an object of a given size, then locate it in your empty canvas prior to the CNN for generation)?

CLEVR:
The images resolution make it hard to really see the shape of the images (here too, the GT would be great). The bounding boxes make the images even harder to parse. I know the colors change but "We can confirm that the model can control both location and object's shape". For the location, it is true, for the shape is hard to completely tell at this resolution without GT.

MS-COCO:
 Just a comment in passing on the fact that resizing images from COCO to 256x256 will inherently distort quite a bit of images, the median size (for each dim) for COCO is 640x480, if I am not mistaken. Most, if not all images in COCO are not 1.0 size ratio.
The quantitative results on COCO seem to confirm that the proposed method is generating "better" images according to IS and FID. This is a good thing, however the technique is strongly supervised (Bounding Box and Object Labels, caption compared to solely captions for StackGAN and AttGAN) so this result should be expected and really put into perspective as your are not comparing models w/ the same supervision (which you mention in the Discussion).

Discussion:
I appreciate that the authors addressed the limitations of their approach in this section. The overlapping BBs seems to be an interesting challenge. Did you try to normalize the embeddings in overlapping area? A simple sum does not seem to be a good solution. In Figure 7 w/ overlapping zebras, the generation seems completely lost.

In terms of clarity, the paper is well-written but would benefit *greatly* from using variables names when discussing 'layout embedding', 'generated local labels', etc. Variable names and equations, while not necessary, can go a long way to clearly express a model's internal blocks (most of the papers you referenced are using this approach). The paper employs none of this commonly used standard and suffers from it. I myself had to write down on the margin the different variables used at each step described in text to have an understanding of what was done (with help of Figure 1). You should reference Figure 1 in the Introduction, as you cover your approach there and the Figure is useful to grasp your contributions.

Another comment concerning clarity is, while it is fine to rely on previously published papers for description of our own work, you should not assume full knowledge from the reader and your paper should stand on its own without having the reader lookup for several papers to have an understanding of your training procedures. If one uses GAN training, it should be expected to cover/formulate quickly the min max game and the various losses you are trying to minimize. I am afraid that "using common GAN procedures" is not enough. When describing your experimental setup, pointing to another paper as "hyperparameters remain the same as in the original training procedure" should not be a substitution for covering it too, even if lightly in the Appendix. For instance: in the Appendix, it is mentioned that training was stopped at 20 epochs for Multi-MNIST, 40 for CLEVR... How did you decide on the epoch (early stopping, stopped before instabillity of GAN training, etc.) Did you use SGD? ADAM? Did you adjust the learning rate, which schedule? etc. for your GAN training. This information in the Appendix would make the paper overall stronger.

Last comment: In terms of generation multiple objects. Have you had the chance to run an object detector on your generated image (you can build one on MSCOCO given the bounding box and label, finetune an ImageNet pretrained model). It would be interesting to see if the generated images are good enough for object detection.

Post-Rebuttal: Given the work from the authors on improving the clarity of the paper as well as investigating the use of object detection metrics to compare their methods, I decided to move my rating upward to 7

---

> ### Author Response · Authors · 2018-11-06
> **Thank you for your detailed comments and feedback (3/3)**
>
> “Another comment concerning clarity is, while it is fine to rely on previously published papers for description of our own work, you should not assume full knowledge from the reader and your paper should stand on its own without having the reader lookup for several papers to have an understanding of your training procedures. If one uses GAN training, it should be expected to cover/formulate quickly the min max game and the various losses you are trying to minimize. I am afraid that "using common GAN procedures" is not enough. When describing your experimental setup, pointing to another paper as "hyperparameters remain the same as in the original training procedure" should not be a substitution for covering it too, even if lightly in the Appendix. For instance: in the Appendix, it is mentioned that training was stopped at 20 epochs for Multi-MNIST, 40 for CLEVR... How did you decide on the epoch (early stopping, stopped before instabillity of GAN training, etc.) Did you use SGD? ADAM? Did you adjust the learning rate, which schedule? etc. for your GAN training. This information in the Appendix would make the paper overall stronger.“
> ----------
> Because of the space constrains we aimed at providing all these information via our Github repository. Hovewer, we will make an effort to also include the information about the general GAN training procedure into our approach section to make the paper more self-contained. Additionally, we will update our appendix to describe the exact training procedure and hyperparameters in more detail.
>
> ----------
> “Last comment: In terms of generation multiple objects. Have you had the chance to run an object detector on your generated image (you can build one on MSCOCO given the bounding box and label, finetune an ImageNet pretrained model). It would be interesting to see if the generated images are good enough for object detection.”
> ----------
> We have not tried this yet. This is an interesting idea and would pose a good additional metric for our paper and the community. We will look into this and keep you updated, whether we can do this in time.
>
> Thank you again for all your valuable comments and your feedback. We will work to implement the feedback we got and will post an updated version of our submission by the end of next week (latest on 16. November) and will let you know once the updated version is online.

---

> ### Author Response · Authors · 2018-11-06
> **Thank you for your detailed comments and feedback (2/3)**
>
> MS-COCO:
> “Just a comment in passing on the fact that resizing images from COCO to 256x256 will inherently distort quite a bit of images, the median size (for each dim) for COCO is 640x480, if I am not mistaken. Most, if not all images in COCO are not 1.0 size ratio.”
> ----------
> We follow the implementation of StackGAN and AttnGAN in preprocessing the images, i.e. we rescale them to 268x268 pixels and then randomly crop a window of size 256x256 pixels. This will indeed distort the images, but we decided to follow the outlined procedure in order to keep our approach more comparable and to not introduce unforeseen effects during training. The implementation is technically not limited to capture any other ratio.
>
> ----------
> “The quantitative results on COCO seem to confirm that the proposed method is generating "better" images according to IS and FID. This is a good thing, however the technique is strongly supervised (Bounding Box and Object Labels, caption compared to solely captions for StackGAN and AttGAN) so this result should be expected and really put into perspective as your are not comparing models w/ the same supervision (which you mention in the Discussion).”
> ----------
> Yes, this is correct. We mention this in the caption of Table 1 and the discussion, but will try to make it even clearer in the updated version.
>
> ----------
> Discussion:
> “The overlapping BBs seems to be an interesting challenge. Did you try to normalize the embeddings in overlapping area? A simple sum does not seem to be a good solution. In Figure 7 w/ overlapping zebras, the generation seems completely lost.”
> ----------
> We did not try any normalization methods for the embeddings in the overlapping area. We leave this open as future work. One feasible approach to normalize the embeddings would be to take the average of all embeddings at a given location, similarly to the Bag-of-Words approach in natural language processing. This would be an easy change in future work and might already improve the performance in the case of overlapping bounding boxes.
>
> ----------
> “In terms of clarity, the paper is well-written but would benefit *greatly* from using variables names when discussing 'layout embedding', 'generated local labels', etc. Variable names and equations, while not necessary, can go a long way to clearly express a model's internal blocks (most of the papers you referenced are using this approach). The paper employs none of this commonly used standard and suffers from it. I myself had to write down on the margin the different variables used at each step described in text to have an understanding of what was done (with help of Figure 1). You should reference Figure 1 in the Introduction, as you cover your approach there and the Figure is useful to grasp your contributions.”
> ----------
> Thank you for adding this view. While writing the paper we considered both a formal and a conceptual description and concluded in favor of latter one. However, we will update our approach section and make our approach clearer by following your suggestions.

---

> ### Author Response · Authors · 2018-11-06
> **Thank you for your detailed comments and feedback (1/3)**
>
> Dear reviewer,
> thank you for your response and detailed feedback. In the following, we reply to your comments sequentially.
>
> ----------
> “The generator Global path is given a noise component. From the text, it does not seem that the Object path is given a noise component. Do you generate always the same object given the same label and Bounding Box then? Why not integrate some noise in this pipeline too?”
> ----------
> As you correctly observed, the object path is not given a noise component. However, there is no reason why integrating noise into the object pathway should not work and this could indeed lead to a higher sample diversity, though we did not test this.
> We observe that not adding any noise to the object pathway does still result in “different” objects for the same label, even for the same caption. This is maybe easiest seen in the Multi-MNIST images (Fig. 2), where in some images the same digit occurs multiple times, but the style is still different for each digit. It can also, to some extent, be observed in the visualization of the object pathway on the MS-COCO data set (Fig. 5, row D). Our hypothesis is, that this is due to the upsampling in the later parts of the generator. Since we concatenate the features of the object and the global pathway (and the global pathway get as input a noise vector) we indirectly introduce the noise component at this point since the generator “merges” the feature representations of the global and the object pathways. Our intuition was to use the object pathway to generate basic, low-level features that are representative for a given object and caption, not to increase the variance of these features for increased object diversity.
>
> ----------
> Multi-MNIST:
> “Could you provide the ground truth labels for each/some image/s? For the failure cases it is often not clear what digit is what. For the Row E and F, 1s could be 7s and vice versa. Since it is a qualitative study, it would be nice to have the Ground Truth (GT) (which you provide to G at for generation).”
> ----------
> We will update our images to also include the ground truth labels for each bounding box to make this clearer.
>
> ----------
> “For the failure case of Row D (right) an interesting results would have been to have example of a digit bounding box from top to bottom with few pixel vertical shift to visualize when the model starts to mess-up the generation. This seems to point that your model (exposed to the location from BB for the object paths) is sensitive to what locations it has seen in training. How would you make the object path more robust to unseen location (overall you need to design an object of a given size, then locate it in your empty canvas prior to the CNN for generation)?”
> ----------
> Thank you, this is a very good suggestion and we will update our submission with examples where we shift the bounding boxes pixel-wise from top to bottom to see when our model starts to fail.
> In order to make the object path more robust to unseen locations, one possibility might be to change the integration between object and global pathways, e.g. by having different object pathways operate at different resolution sizes. Since the issue with unseen locations seems to be the upsampling after the concatenation of object and global pathways the issue might be somewhat alleviated by incorporating object pathways that also work on higher parts of the generator, i.e. greater resolutions. However, we also observe that the problem with unseen locations seems to be at its worst when some locations are not observed “at all” during training (independent of the object class, i.e. if no object at all has been observed at a given location during training). As we can see in row D (left) of Fig. 2, as long as some objects are seen at a location the generator seems to be able to generalize to other objects at that location during test time. As such we speculate that the issue with unseen locations might not be crucial in real-world data, as long as we have enough data so that some kind of object is observed during training in all different locations of the images. As such, we suspect that the approach is quite robust as long as we have a somewhat balanced training set in the sense that object locations are not localized to specific areas within the image.
>
> ----------
> CLEVR:
> “The images resolution make it hard to really see the shape of the images (here too, the GT would be great). The bounding boxes make the images even harder to parse. I know the colors change but "We can confirm that the model can control both location and object's shape". For the location, it is true, for the shape is hard to completely tell at this resolution without GT. “
> ----------
> We will also update the figures of the CLEVR data set to include the ground truth labels for each bounding box in the updated submission.

---

### Official Review · AnonReviewer2 · 2018-11-03
**Simple but interesting idea for controling image generation**

**Rating:** 8
**Confidence:** 4

**Review:**

The paper proposes a simple but effective method for controlling the location of objects in image generation using generative adversarial networks. Experiments on MNIST and CLEVR are toy examples but illustrate that the model is indeed performing as expected. The experiments on COCO produce results that while containing obvious artefacts are producing output consistent with the input control signal (i.e., bounding boxes). It would however have been interesting to see more varied bounding box locations for the same caption.

In short, the paper makes an interesting addition to image generation works and likely to be incorporated into future image generation and inpainting methods.

---

> ### Author Response · Authors · 2018-11-06
> **Thank you for your kind feedback and comments**
>
> Dear reviewer,
> thank you very much for your review.
>
> We will update our submission with examples of images based on MS-COCO captions in which we vary the location of the various bounding boxes.
>
> We will work to implement the feedback we got and will post an updated version of our submission by the end of next week (latest on 16. November) and will let you know once the updated version is online.

---

### Official Review · AnonReviewer3 · 2018-11-04

**Rating:** 6
**Confidence:** 4

**Review:**

This paper proposed a model to generate location-controllable images built upon GANs. The experiments are conducted on several datasets. Although this  problem seems interesting, here are several concerns I have:

1.Novelty: the overall framework is still conditional GAN framework. The multiple -generators-discriminators structure has been used in many other works (see the references). The global-local design is not new. Finally, compared with Reed et al. [2016], the novelty is limit.

2.Motivation: I still can not tell why the proposed method is better than ones with scene layout. For me, the cost of collecting annotated data is almost the same.

3. The experimental results are week.  For such a task, it is difficult to find a good metric. Thus the qualitative comparison is important. I think the author should follow standard rule to do some design for user study instead of cherry pick some examples. Besides, it should include more baselines instead of StackGAN.

References:
a. Xi et al. Pedestrian-Synthesis-GAN: Generating Pedestrian Data in Real Scene and Beyond
b. Yixiao et al. FD-GAN: Pose-guided Feature Distilling GAN for Robust Person Re-identification

Revision:
Thanks for the work of the authors' and all the reviewers. I spent sometime reading the rebuttal as well as the revised paper. It addressed most of my concern. I would like to change my rating from 5 to 6.

---

> ### Author Response · Authors · 2018-11-06
> **Thank you for your helpful feedback and comments (2/2)**
>
> ----------
> "2.Motivation: I still can not tell why the proposed method is better than ones with scene layout. For me, the cost of collecting annotated data is almost the same."
> ----------
> Thank you for pointing to the lack of clarity here. In fact we think that using semantic scene layouts it not necessarily worse but just yields different properties. Our approach requires both the bounding boxes and the associated labels but arguably "less" information than an image scene layout (bounding box level annotation versus pixel level annotation). Nevertheless, we agree that the cost of collecting the required data may be similar.
> On an intuitive level, if we want a human to generate an image we would usually only give them a general description (image caption) and possibly the location where we want the salient objects to be. This is essentially what we do with our model, as opposed to describing "common sense" knowledge in detail such as "the sky should be in the top of the image" and "the grass should be at the bottom", which is essentially what semantic scene layouts do.
> Another advantage of using bounding boxes instead of semantic layout is that they make it easier for humans to manually change the layout of the scene (semantic scene manipulation) in potential downstream tasks since humans usually do this on a per-object basis and not on a per-pixel basis.
>
> ----------
> “3. The experimental results are week.  For such a task, it is difficult to find a good metric. Thus the qualitative comparison is important. I think the author should follow standard rule to do some design for user study instead of cherry pick some examples. Besides, it should include more baselines instead of StackGAN.”
> ----------
> We agree that a user study would be a valuable next step for all generative approaches within the community, but is often difficult to do because of time and resources constraints. To reduce the tendency of cherry picking, we made an effort to include both: well-working examples as well as failure cases (see e.g. Fig 2 rows D-F and last three examples of Fig. 6 and Fig. 7 respectively).
> Additionally, we report the IS and FID values in order to provide results that are comparable with related models (see Table 1). As a direct baseline for qualitatively comparing the images on the MS-COCO data set we used both the StackGAN since it is a well-known architecture that performs well on many different data sets and the AttnGAN since it provides the current SOTA in image generation on the MS-COCO data set (based on IS score). We acknowledge that other interesting approaches are emerging e.g. from CVPR (e.g. [1-3]) and we will make an effort to provide further tests if the implementation and training are feasible in time (e.g. [1] and [2]).
>
> Overall, we would like to thank you again for your valuable feedback and concerns. We will work to implement the feedback we got and will post an updated version of our submission by the end of next week (latest on 16. November) and will let you know once the updated version is online.
>
> [1] Photographic Text-to-Image Synthesis with a Hierarchically-nested Adversarial Network, Zhang Zizhao et al, CVPR, 2018
> [2] Image Generation from Scene Graphs, Justin Johnson et al, CVPR, 2018
> [3] Inferring Semantic Layout for Hierarchical Text-to-Image Synthesis, Seunghoon Hong et al, CVPR, 2018

---

> ### Author Response · Authors · 2018-11-06
> **Thank you for your helpful feedback and comments (1/2)**
>
> Dear reviewer,
> thank you very much for your feedback. In the following, we reply to your concerns one by one.
>
> ----------
> "1. Novelty: the overall framework is still conditional GAN framework. The multiple -generators-discriminators structure has been used in many other works (see the references).”
> ----------
> It is true that our overall framework falls into the class of conditional GANs, as do most other GAN frameworks that aim to gain more control over the image generation process.
>
> However, our model itself does not require multiple generators or multiple discriminators. In principle, the object pathway can be added to any GAN architecture (e.g. DCGAN) and is not reliant on multiple generators or discriminators, but instead augments existing generators/discriminators. Note, for example, that for our experiments on the Multi-MNIST and the CLEVR data sets we only use one generator and one discriminator. For our experiments on the MS-COCO data set we extend two common GAN architectures (StackGAN and AttnGAN) that are indeed composed of multiple generators and discriminators. However, as noted, this is not a requirement of our model but simply happened to be a characteristic of the baseline architectures we chose to use for the MS-COCO data set, based on their excellent performance.
> Thank you pointing to the brand new references. There is interesting progress in the upcoming NIPS and we will make an effort to include them in our related work section where appropriate.
>
> ----------
> “The global-local design is not new. Finally, compared with Reed et al. [2016], the novelty is limit."
> ----------
> The global-local framework is indeed used by Reed et al. (as we mention in our related work Reed et al.'s work is closely related to ours). In the following we summarize the key architectural differences between our work (focus on multiple objects per image) and Reed et al. (focus on one object per image).
> On the generator side, Reed et al. spatially replicate the image caption at the location of the bounding box and then uses a CNN to encode this information. In contrast, we first use a dense layer, which gets as input the image caption embedding and the local bounding box label as a one-hot encoding, to obtain a new localized label which is replicated at the respective bounding box location. This step is repeated for each object in the image (the same dense layer is used to obtain each object label) and results in multiple labels which are replicated at their respective bounding box locations leading to our layout encoding which is used by the global pathway. Reed et al.’s local pathway generated image features of one centralized object. In contrast, we apply our object (local) pathway multiple times based on the previously generated labels and generate feature representations of each of the objects at the locations specified by their bounding boxes.
> On the discriminator, Reed et al. global pathway is similar to ours. Their local pathway, however, first downsamples the full image, then concatenates it with the image caption, then crops the representation to the location of the bounding box, and then applies more convolutional layers to obtain the object features. In contrast, our object (local) pathway is applied iteratively and gets as input the image content directly (RGB values) at the location of each of the bounding boxes (not the whole image), concatenated with the respective label of that bounding box (one-hot encoding). As such, the output of our object (local) pathway are localized image features at the locations of the bounding boxes based on the image content and bounding box labels at exactly these locations.
> We hope we could clarify some of the key differences between our architecture and the architecture by Reed et al. We will also make these differences clearer in the related work section of our updated submission.

---

### Author Response · Authors · 2018-11-16
**Updated Revision**

Dear reviewers,

thanks to your helpful feedback and replies we were able to improve our submission and we just uploaded the updated version of our paper.

In the main part of the paper we made the following improvements (we mention only the major points):
    - Related Work: we updated the related work section to highlight the differences between our work and the work by Reed et al. in more detail.
    - Approach: we made the section more self-contained, explained the GAN training procedure in more detail, described (on a high level) the general objective function that we optimize, and included formal descriptions for various parts of our model (also reflected in Fig. 1).
    - We updated Fig. 2 in a way that it now also shows the ground truth labels for all bounding boxes.

In the appendix we updated the following parts:
    - Implementation details: we describe the implementation details and all hyperparameters for all experiments in much more detail.
    - We added a figure (Fig. 6) detailing the failure cases in Fig 2. row D (right), where we move the bounding boxes iteratively from the top to the bottom of the image to study when the model breaks in generating recognizable digits.
    - We added two figures (Fig. 8 + Fig. 9) showing more variations in the location of the bounding boxes (including failure cases) on the MS-COCO data set, both for the StackGAN and the AttnGAN architecture.

We think that these changes further improve the overall quality of our submission.
Thanks again to all reviewers for their helpful comments.

---

### Author Response · Authors · 2018-11-26
**Final Revision Uploaded**

Dear reviewers,

we have uploaded a final revision of our paper.
As suggested by AnonReviewer1 we additionally evaluated the performance of our generative models by running an object detector (YOLOv3) on the generated images.

We selected the 30 most common labels from the MS-COCO data set (based on how often these labels occur in the captions of the test set) and generate images for each label-caption pair and model. On these images, we measured the recall, i.e. how often the YOLOv3 network detects the given object, and calculate the Intersection over Union (IoU) between detected objects and ground truth for the images.

The results confirm our previous observations:
- Using the object pathway results in the YOLOv3 network detecting the given object more often, regardless of the used model (StackGAN or AttnGAN), indicating an increased image quality.
- The StackGAN achieves a comparably high IoU (greater than 0.3 for all tested labels and greater than 0.5 for 86.7% of the tested labels).
- The images from the AttnGAN seem to be of higher quality than images generated by the StackGAN, which leads to an even higher detection rate of the given objects by the YOLOv3 network; however, the average IoU is smaller than for the StackGAN (only 53.3% of the tested labels have an IoU greater than 0.5), probably due to the fact that the AttnGAN tends to place features of salient objects at many locations throughout the image (also observed in the other experiments).

The detailed results and exact methodology have been added to the paper as well.

We think that this additional evaluation further strengthens the results of our paper and highlights the advantages of using an object pathway in the GAN framework.

We thank all reviewers again for their valuable feedback and comments which greatly helped to improve the quality of our submission.

---

### Author Response · Authors · 2019-01-04
**Camera Ready Paper Uploaded**

Dear reviewers and readers,

we have uploaded the camera ready, de-anonymized version of our paper.
The code, data, and models to reproduce the paper's results can be found here: https://github.com/tohinz/multiple-objects-gan

---

### Meta-Review · Area_Chair1 · 2018-12-14

**Confidence:** 4
**Recommendation:** Accept (Poster)

**Metareview:**

The submission proposes a model to generate images where one can control the fine-grained locations of objects. This is achieved by adding an "object pathway" to the GAN architecture. Experiments on a number of baselines are performed, including a number of reviewer-suggested metrics that were added post-rebuttal.

The method needs bounding boxes of the objects to be placed (and labels). The proposed method is simple and likely novel and I like the evaluating done with Yolov3 to get a sense of the object detection performance on the generated images. I find the results (qual & quant) and write-up compelling and I think that the method will be of practical relevance, especially in creative applications.

Because of this, I recommend acceptance.